# Redox-induced controllable engineering of MnO$_2$-Mn$_x$Co$_{3-x}$O$_4$ interface to boost catalytic oxidation of ethane

Haiyan Wang[1,6], Shuang Wang[1,6], Shida Liu[2,6,7] ✉, Yiling Dai[3], Zhenghao Jia [®][4], Xuejing Li[2], Shuhe Liu[2], Feixiong Dang[1], Kevin J. Smith[5], Xiaowa Nie [®][1,7] ✉, Shuandi Hou [®][2,7] ✉ & Xinwen Guo [®][1,7] ✉

Multicomponent oxides are intriguing materials in heterogeneous catalysis, and the interface between various components often plays an essential role in oxidations. However, the underlying principles of how the hetero-interface affects the catalytic process remain largely unexplored. Here we report a unique structure design of MnCoO$_x$ catalysts by chemical reduction, specifically for ethane oxidation. Part of the Mn ions incorporates with Co oxides to form spinel Mn$_x$Co$_{3-x}$O$_4$, while the rests stay as MnO$_2$ domains to create the MnO$_2$-Mn$_x$Co$_{3-x}$O$_4$ interface. MnCoO$_x$ with Mn/Co ratio of 0.5 exhibits an excellent activity and stability up to 1000 h under humid conditions. The synergistic effects between MnO$_2$ and Mn$_x$Co$_{3-x}$O$_4$ are elucidated, in which the C$_2$H$_6$ tends to be adsorbed on the interfacial Co sites and subsequently break the C-H bonds on the reactive lattice O of MnO$_2$ layer. Findings from this study provide valuable insights for the rational design of efficient catalysts for alkane combustion.

Low-chain alkanes (C$_1$-C$_3$) that released from industrial processes, have gained considerable attention because of the growing concerns regarding air quality and human health[1,2]. Of particular interest in ethane, a representative non-methane volatile organic compounds (NMVOCs), has become the focus of regulatory scrutiny due to stringent standards on flue gas emissions[3]. As a result, various technologies have been developed to mitigate their emission. Catalytic oxidation is shown to be an effective approach for eliminating alkanes and their derivatives[4]. However, the development of high-performance catalysts, particularly for low temperature application, remains challenging due to the inherently strong C-H bonds. Additionally, ethane that vastly exists in nature gas (1–9 mol%), must be taken into account during the catalytic nature gas combustion to examine its energetic

performance on combustion process as well as its impact on the employed materials. Noble metal-based catalysts (such as Pt or Pd) typically exhibit excellent catalytic activity toward low-chain alkane combustion at low temperature[5]. However, the high cost and limited availability have driven people to find alternatives. With this goal in mind, a substantial amount of work was undertaken to develop efficient non-noble metal-based catalysts[6].

Special attention has been given to transition metal spinel-type oxides (AB$_2$O$_4$) because of their remarkable activities and durability in oxidation reactions[7–9]. In a typical AB$_2$O$_4$ structure, the tetrahedral and octahedral sites are occupied by A$^{2+}$ and B$^{3+}$ cations, respectively, offering a unique atomic arrangement that allows the facile tuning of the redox property[10,11]. Additionally, the interaction between A and B in

[1]State Key Laboratory of Fine Chemicals, Frontiers Science Center for Smart Materials, School of Chemical Engineering, Dalian University of Technology, Dalian 116024, P.R. China. [2]SINOPEC Dalian (Fushun) Research Institute of Petroleum and Petrochemicals, Dalian 116045, P.R. China. [3]Key Lab of Organic Optoelectronics & Molecular Engineering, Department of Chemistry, Tsinghua University, Beijing 100084, China. [4]Division of Energy Research Resources, Dalian Institute of Chemical Physics, Chinese Academy of Sciences, 457 Zhongshan Road, Dalian 116023, China. [5]Department of Chemical and Biological Engineering, University of British Columbia, 2360 East Mall, Vancouver, B.C. V6T 1Z3, Canada. [6]These authors contributed equally: Haiyan Wang, Shuang Wang, Shida Liu. [7]These authors jointly supervised this work: Shida Liu, Xiaowa Nie, Shuandi Hou, Xinwen Guo. ✉e-mail: liushida.fshy@sinopec.com; niexiaowa@dlut.edu.cn; houshuandi.fshy@sinopec.com; guoxw@dlut.edu.cn

spinels was identified, which accelerates the generation of reactive oxygen[12]. Moreover, the electron configuration of metal ions in $AB_2O_4$ spinels can be readily tuned by metal doping, thereby alternating the adsorption strength of reactants[13]. However, it should be noted that the obtained spinel oxides may not always present in an ideal $AB_2O_4$ structure, because various factors are involved in the synthesis process[14]. In some cases, certain amount of metal ions may become isolated from their parent spinel grains during crystal growing or post-synthesis process, resulting in the formation of multi-phase oxides. The properties of mixed oxides are more complex than that of the pure spinel because of the involved various interfaces and their coordination environments. Therefore, further investigation is required to elucidate the inherent properties of these interfacial sites as well as optimize the synthesis parameters, to achieve better control over the microstructure of synthesized catalysts.

Interfacial engineering has emerged as an important approach in the design of first catalytic materials, enabling the facilitation of diverse chemical reactions, such as dehydrogenation[15], CO oxidation[16,17], and water-gas shift reaction[18,19]. As stimulated by the growing interests in interface catalysis, extensive investigations have been conducted to understand the properties of active sites located within the heterojunction region of mixed oxides. Notably, it has been observed that the interface of multicomponent oxides plays a vital role in facilitating the mass transfer of oxygen. Zhu et al.[20] found that the proximity between $MnO_2$ and $CeO_2$ increased the mobility of both surface and lattice oxygen around the grain boundary of $MnO_2$-$CeO_2$ interface, resulting in an enhanced activity in HCHO removal ($T_{100} = 100\,°C$, GHSV = 90 L h$^{-1}$). Similarly, Zhang et al.[21] optimized the structure of $ZnCo_2O_4@CeO_2$ catalyst, and discovered that the nanoscale contacts between $ZnCo_2O_4$ and $CeO_2$ introduce an enhanced oxygen storage capacity and lattice oxygen mobility. Shan et al.[22] adopted the acid-etching approach to create $MnO_2$-$CoMn_2O_4$ interfacial system and unveiled that the lattice O that located at interfacial sites was activated due to the weakened Mn-O bonds as well as the altered coordination environments of O atoms. Also, Ren et al.[23] discovered that the concentration of oxygen vacancies of $CoMn_2O_4$ spinel significantly increased after $HNO_3$ treatment, therefore generating more active surface O species during $O_2$ activation. Likewise, the established $CeO_2$-$Co_3O_4$ interface in $CeO_2@Co_3O_4$ nanofiber catalysts has proved to be effective in propane oxidation[24]. Moreover, the intimate contact between mixed oxides gives rise to a synergistic catalytic effect, allowing for the simultaneous activation of different reactants. Zhu et al.[25] investigated the dual interfacial effects between PtFe and $FeO_x$ in each nanowire (NWs) as well as the interaction between NWs and $TiO_2$ support on the PtFe-$FeO_x$/$TiO_2$ catalyst, and discovered their interfacial synergy in CO oxidation. Liu et al.[26] designed a hierarchical $MnO_2@NiCo_2O_4@Ni$ foam catalyst, and found that the three-dimensional core-shell structure maximizes the interaction between $NiCo_2O_4$ and $MnO_2$, consequently improving the performance of $NH_3$-SCR at low temperature. Zhang et al.[27] constructed $AgO/CeSnO_x$ tandem catalysts and studied the synergistic effects between AgO and $CeSnO_x$ dual sites for selective oxidation of $NH_3$. It is noticed that the electrons on $CeSnO_x$ support were more easily transferred to AgO NPs, which accelerates the oxidation activity of AgO and the reduction performance of $CeSnO_x$ support, thus achieving a good match between $NH_3$ oxidation and $NO_x$ reduction. Also, the strong interaction between different metal oxides affects the dispersion and crystallinity of active centers[28,29]. Furthermore, the electronic property at interfacial region could be flexibly altered by tuning the interaction between different components[30]. These examples emphasize the crucial role of interfaces in multicomponent catalysts. Hence, it is imperative to find out how critical the formed interfaces dictate the performance of complex oxides and further obtain a fundamental understanding on the "property-activity" relationship.

Herein, we report our finding on the manipulation of the $MnO_2$-$Mn_xCo_{3-x}O_4$ interface through engineering the Mn/Co ratio of $MnCoO_x$ catalysts or adjusting the annealing conditions. The resulting materials predominantly exhibit as $Mn_xCo_{3-x}O_4$ spinel oxides with $MnO_2$ thin layers decorated on the surface. The catalytic performance of the obtained $MnCoO_x$ catalysts was evaluated in ethane oxidation to build the correlation between their structure and performance. Our results showed that the presence of $MnO_2$ and $Mn_xCo_{3-x}O_4$ at the grain boundary of the involved oxides synergistically enhanced both the ethane adsorption/activation and the lattice oxygen mobility. Notably, the strong interaction between $MnO_2$ and $Mn_xCo_{3-x}O_4$ induces a charge rearrangement between these components as supported by the in-situ X-ray Photoelectron Spectroscopy (XPS) analysis and Density Functional Theory (DFT) calculations. Elucidating the role of structural heterogeneity in multicomponent oxides enables us to selectively tune the interface properties and oxygen defects in a wide range of complex oxides.

## Results
### Structure and surface states
A series of Mn-substituted cobalt oxides ($MnCoO_x$-z with varied Mn/Co ratios (z) of 0–2.0) were successfully prepared by chemical reduction method. The obtained $MnCoO_x$ catalysts present as hierarchical nanospheres with an average diameter of 250-500 nm, which is mainly composed by ultrathin nanosheets with the surface covered by thin layers (Fig. 1a, b; Supplementary Fig. 1). A schematic illustration is presented to show the formed grain boundary layers as a function of Mn/Co ratio (Fig. 1c).

Firstly, the evolution of composition-dependent crystal structure of $MnCoO_x$ catalysts was examined. Figure 1d presents the Raman spectra of $MnCoO_x$ catalysts. Note that, the Raman spectra of $MnCoO_x$-0.1 is similar to that of $Co_3O_4$ reference. While, the main peak of octahedrally coordinated Co sites ($CoO_6$: 670 cm$^{-1}$) gradually shifted to lower wavenumber and merged with the shoulder peak (604 cm$^{-1}$) to form a broader peak when Mn/Co ratio is ≥ 0.2, implying the weakened vibration of Co-O bonds. Similar phenomenon was also observed in $Ni_xCo_{3-x}O_4$ spinel[31]. Also, the added Mn ions significant altered the symmetry of $CoO_6$, resulting from the lattice replacement induced inhomogeneous distribution of Mn(III) or Co(III/II) ions[32–34]. The induced coordination environmental change further initiates the occurrence of structural defects and lattice distortion on the developed $MnCoO_x$, which in turn benefits the formation of oxygen vacancies. Besides, the peak position of tetrahedrally coordinated Co ($CoO_4$: 191 cm$^{-1}$) was invariant with varied Mn/Co ratio, but their intensity decreased at high Mn/Co ratio due to Mn substitution. Similar result was also obtained from FT-IR analyses (Supplementary Fig. 2). Meanwhile, no active Raman bands belong to Mn-O bonds (as indicated by the blue dash line in Fig. 1d) were observed in the prepared $MnCoO_x$ catalysts, suggesting that the Mn ions are highly dispersed and/or exist as solid solution in $Co_3O_4$. The bulk structure of $MnCoO_x$ was further studied by power X-ray diffraction (XRD) (Supplementary Fig. 3, Supplementary Table 1). The results indicate the incorporation of Mn ions into $Co_3O_4$ lattice, leading to the formation of $MnCo_2O_4$ spinel (PDF#23-1237). Also, the selected area electron diffraction (SAED) pattern (the insert of Fig. 1b) is indexed to the cubic lattice typical of $MnCo_2O_4$.

To get more insights of Mn species, X-ray photoelectron spectroscopy (XPS) measurements were performed to investigate the surface states of Mn-O-Co entity (Fig. 1e). Clearly, the surface atomic ratios of Mn/Co measured by XPS (Supplementary Table 2) were higher than that of the corresponding bulk Mn/Co ratio measured by ICP-OES, indicating that part of the Mn ions was dispersed on the surface of $MnCoO_x$ catalysts. Notably, the $MnCoO_x$ presented a high $Co^{2+}/Co^{3+}$ ratio of 0.4 ~ 0.6 compared to $Co_3O_4$ ($Co^{2+}/Co^{3+} = 0.35$,

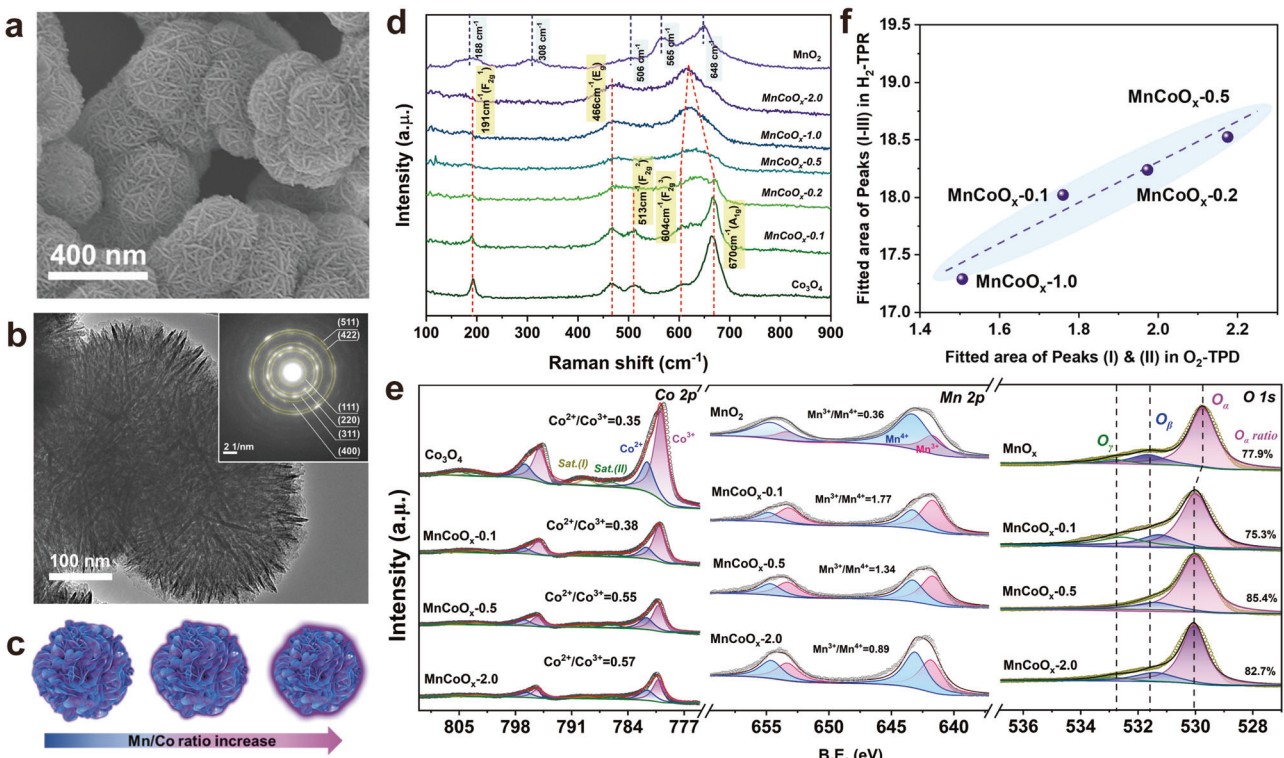

**Fig. 1 | Structural analyses of the as-synthesized MnCoO$_x$ catalysts. a** SEM image of MnCoO$_x$-0.5. **b** TEM image of MnCoO$_x$-0.5 with an insert showing a corresponding electron diffraction (SAED) pattern. **c** Schematic illustration of the grain boundary of MnCoO$_x$ with varied Mn/Co ratio. **d** Raman spectra. (Yellow shading area: the vibrational bonds of Co species in MnCoO$_x$; blue shading area: the vibrational bonds of Mn species in MnO$_2$) **e** XPS spectra. **f** A correlation of cumulative area under H$_2$ reduction peaks (I & II & III) and O$_2$ desorption peaks (I & II) (Dash line: it was drawn to guide the readers' eyes). (Source Data are provided as a Source Data file).

Supplementary Table 3), an indicator of Mn substitution into the octahedral sites of Co$^{3+}$. Also, the presence of satellite peaks suggests the partial reduction from Co$^{3+}$ to Co$^{2+}$, which demonstrates the coexistence of Co$^{2+}$ and Co$^{3+}$ on the prepared MnCoO$_x$ catalysts[35–37]. The Co$^{2+}$/Co$^{3+}$ ratio increases with increased Mn addition and levels off above Mn/Co of 0.5. The formed Mn$^{3+}$ ions increase the anionic defects as Co or Ni does in other spinels, thus benefiting the catalytic oxidation process[38–40]. From Mn $2p_{3/2}$ spectra, we can notice that the MnCoO$_x$-0.1 catalyst showed the highest Mn$^{3+}$/Mn$^{4+}$ ratio, indicating that more O$_v$ are created to maintain the electrostatic balance of the system ($4Mn^{4+}+O^{2-}\rightarrow2Mn^{4+}+2Mn^{3+}+\square+0.5O_2$)[41–43]. A gradual decrease of Mn$^{3+}$/Mn$^{4+}$ ratio appeared while increasing the Mn/Co ratio due to the diffusion of MnO$_2$ onto the surface of Mn$_x$Co$_{3-x}$O$_4$ substrate. The average oxidation state (AOS) of Mn 3 s increased with increasing the Mn/Co ratio, which is consistent with Mn $2p$ results (Supplementary Fig. 4). Therefore, we can infer that the added Mn mainly remain in two states, in which part of the Mn is incorporated into the bulk structure of Co$_3$O$_4$ to form Mn$_x$Co$_{3-x}$O$_4$ spinel, and the rest contributes to the formation of MnO$_2$ layer or aggregates as determined by the amount of added Mn.

Moreover, the O 1$s$ spectra were fitted into three peaks, which attributed to lattice oxygen (O$_\alpha$), surface adsorbed oxygen (or defects, O$_\beta$), and chemisorbed water (O$_\gamma$) with B.E.s of 530.1, 531.3, and 532.7 eV, respectively[44,45]. The O$_\alpha$ species account about 75–85% over MnCoO$_x$ catalysts, indicating their significant role in oxidation reaction. In addition, O$_2$-TPD (Supplementary Fig. 5a) was performed to study the type and mobility of oxygen that contained in the MnCoO$_x$ catalysts. It was found that the O$_2$ desorption peak in the range of 300–600 °C (Region II) obviously shifted towards low temperature on MnCoO$_x$ catalysts compared to MnO$_2$ and Co$_3$O$_4$ references, implying an improved oxygen mobility after Mn addition. However, the desorption

amount in Region II dramatically decreased when Mn/Co ratio is above 0.5, perhaps due to the excessive accumulation of MnO$_2$ on the surface. This trend is consistent with what we observed from EPR analysis (Supplementary Fig. 6), indicating that there was more O$_v$ on MnCoO$_x$-0.5.

To clarify the reducibility of involved oxides and the interaction of various species, the H$_2$ reduction peak was roughly divided into three individual peaks for MnCoO$_x$ catalysts with Mn/Co ratio of 0.1–0.5 (Supplementary Fig. 5b). Peak(I) appearing at 100–200 °C belongs to the surface adsorbed O[39]. Noted that the peak (I) accounts for 20% of all the consumed H$_2$ on the MnCoO$_x$-0.1 catalyst, while this value decreased to 10% once more Mn was introduced (Supplementary Tables 2 and 3). The relative amount of peak (II) increased with increased Mn/Co ratio (max.26%), indicating the appearance of MnO$_2$ on the surface of MnCoO$_x$. Also, it is noticeable that the reduction peak (III) shifted towards the lower temperature region (355–375 °C) compared to the bulk Co$_3$O$_4$ (387 °C), perhaps due to the facile H$_2$ transfer from MnO$_2$-Mn$_x$Co$_{3-x}$O$_4$ interface to the bulk materials. Similar phenomenon was also observed on Mn$_2$O$_3$@MnO$_2$ catalyst via MnO$_2$-Mn$_2$O$_3$ interface[45]. Note that the total integrated area of peaks (I) and (II) in the O$_2$-TPD analysis exhibits a linear correlation with the cumulative area under H$_2$ reduction obtained from (I), (II), and (III) peaks (Fig. 1f, Supplementary Table 4). However, the excessive amount of Mn shifts peak (III) towards high temperature and even induces the formation of peak (IV), a suggestive of the strong interaction between Mn and Co oxides[12]. To better understand the low-temperature reducibility of MnCoO$_x$ catalysts, the initial H$_2$ consumption rate was calculated and plotted as a function of inversed temperature (1/T), as shown in Supplementary Fig. 7a. Clearly, the initial H$_2$ consumption rate decreased in the sequence of MnCoO$_x$-0.5 > MnCoO$_x$-0.2 > MnCoO$_x$-0.1 > MnCoO$_x$-1.0.

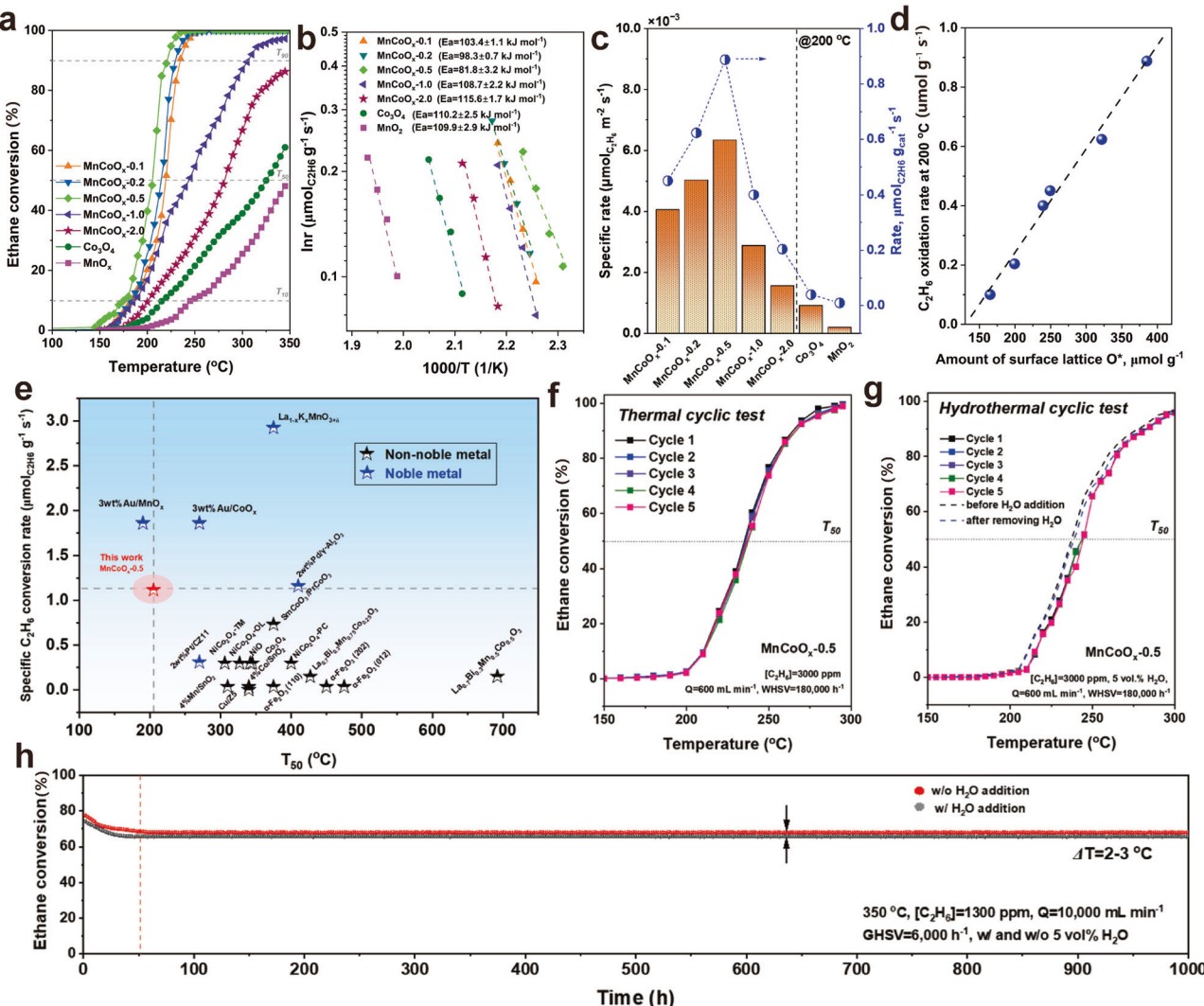

**Fig. 2 | Catalytic performance of MnCoO$_x$-0.5 catalysts for ethane oxidation.**
**a** Light-off curves of the as-prepared catalysts (reaction conditions: *ca.* 200 mg catalyst, [C$_2$H$_6$] = 3000 ppm, Q = 200 mL min$^{-1}$ and WHSV = 60,000 h$^{-1}$).
**b** corresponding Arrhenius plots. **c** specific ethane conversion rate at 200 °C. **d** a correlation of the fitted peak area of oxygen species from O$_2$-TPD analysis with C$_2$H$_6$ oxidation rate at 200 °C. **e** comparison of ethane conversion rate at T$_{50}$ with other catalysts reported in the literature (see Table S9 for details). **f** cyclic thermal

stability test of MnCoO$_x$-0.5 catalyst. **g** cyclic hydrothermal stability test of MnCoO$_x$-0.5 catalyst (reaction conditions: *ca.* 200 mg catalyst, [C$_2$H$_6$] = 3000 ppm, Q = 600 mL min$^{-1}$, WHSV = 180,000 h$^{-1}$ w/o and w/ 5 vol% H$_2$O, respectively). **h** long-term scale up stability tests by 1 wt% MnCoO$_x$-0.5 coated on micro-monolith substrate for 1000 h (reaction conditions: 350 °C, [C$_2$H$_6$] = 1300 ppm, Q = 10,000 mL min$^{-1}$, and GHSV = 6000 h$^{-1}$, w/ and w/o 5 vol% of H$_2$O). (Source Data are provided as a Source Data file).

## Catalytic performance evaluation

To determine the influence of Mn addition, all the synthesized MnCoO$_x$ catalysts were employed for ethane combustion (Fig. 2a, Supplementary Table 5). Taking the temperature at 50% ethane conversion (T$_{50}$) as an indicator, we found that the oxidation activities decreased in the order of MnCoO$_x$-0.5 (205 °C) > MnCoO$_x$-0.2 (215 °C) > MnCoO$_x$-0.1 (219 °C) > MnCoO$_x$-1.0 (260 °C) > MnCoO$_x$-2.0 (282 °C) > Co$_3$O$_4$ (325 °C) > MnO$_2$ (348 °C), suggesting that a small amount of Mn can greatly enhance the activity. However, the activity is sluggish while adding too much Mn, perhaps due to the aggregation of MnO$_2$. To further evaluate the commercial potential of MnCoO$_x$ catalysts, the catalytic activities of other low-chain alkanes (CH$_4$ and C$_3$H$_8$) were tested since they are also contained in the industrial emission (Supplementary Fig. 8, Tables 6–7). It is well-known that the initial H abstraction of short-chain alkanes is often regarded as the key elementary step[7,46]. The strength of C-H bond is closely related to the chain length of alkanes, which in turn determines their reactivity in oxidation reactions. Specifically, the C-H bonds become weaker as the

chain length increased (1$^{st}$ C-H bond strength: CH$_4$ (465 kJ mol$^{-1}$) > C$_2$H$_6$ (442 kJ mol$^{-1}$) > C$_3$H$_8$ (427 kJ mol$^{-1}$))[47]. For comparison, the catalytic activity of Co-based oxides for low-chain alkane (C$_1$-C$_3$) combustion were summarized in Supplementary Table 8. Clearly, the prepared MnCoO$_x$-0.5 in this work revealed a better catalytic activity than many of the reported catalysts in the literature.

To further evaluate the catalytic performance of MnCoO$_x$ catalysts, a kinetic study was completed. Figure 2b presents the Arrhenius plots of MnCoO$_x$ catalysts for ethane combustion based on the normalized reaction rates at ethane conversion in the range of 5-10%. The obtained apparent activation energy (E$_a$) of MnCoO$_x$ is in the range of 80-116 kJ mol$^{-1}$, exhibiting a volcano-typed trend with increased Mn content. Also, the calculated E$_a$ is strongly correlated with the reactivity of MnCoO$_x$ catalysts. Note that, the E$_a$ value of MnCoO$_x$-0.5 catalyst (E$_a$ = 81.8 ± 3.2 kJ mol$^{-1}$) is the lowest, indicating an easier oxidation of C$_2$H$_6$. Also, the turn-over frequency (TOF) of MnCoO$_x$-0.5 catalyst for ethane oxidation is 3.93 × 10$^{-2}$ s$^{-1}$ at 200 °C, which is significantly higher than other MnCoO$_x$ samples. A good correlation was

built between TOF and the initial $H_2$ consumption rate for $MnCoO_x$ catalysts, as shown in Supplementary Fig. 7b. These results suggest that the $MnCoO_x$-0.5 catalyst with low $E_a$ (81.8 ± 3.2 kJ mol$^{-1}$) and high TOF (3.93 × 10$^{-2}$ s$^{-1}$) is more effective for ethane oxidation on per site basis. Moreover, the effect of space velocity on catalytic activity of $MnCoO_x$-0.5 catalyst was investigated as shown in Supplementary Fig. 9. Clearly, the ethane conversion decreased with the increased WHSV, as a result of shortening the contact time.

To study the intrinsic activity of $MnCoO_x$ catalysts, the areal rates normalized by the specific surface area (Supplementary Fig. 10, Supplementary Table 9) of synthesized catalysts (expressed in the unit of $\mu$mol m$^{-2}$ s$^{-1}$) were calculated and plotted in Fig. 2c. $MnCoO_x$-0.5 catalyst showed the highest areal rate (6.3 ×10$^{-3}$ $\mu$mol m$^{-2}$ s$^{-1}$), which might be attributed to the strong chemical interaction between Mn and Co oxides, thus creating more effective interfacial sites and further changes the interaction between reactants and lattice O upon Mn substitution. The specific ethane oxidation rate either as per surface area or per mass of prepared catalysts exhibited a similar volcano-typed trend as a function of Mn/Co ratio. This trend is in good agreement with the calculated $E_a$. Also, a linear correlation was established between $C_2H_6$ oxidation rate and the amount of surface or subsurface lattice oxygen species, as calculated by the cumulative area of peak (II) in $O_2$-TPD results (Fig. 2d). Note that the prepared $MnCoO_x$-0.5 catalyst showed a superior catalytic performance in ethane oxidation compared to the reported non-noble metal catalysts so far, and even better than several reported noble-metal supported catalysts (Fig. 2e, Supplementary Table 10).

Moreover, the cyclic stability tests were performed both under dry and humid conditions at a relatively high WHSV of 180,000 h$^{-1}$ (Fig. 2f, g, Supplementary Fig. 11). As shown in Fig. 2f, the $MnCoO_x$-0.5 catalyst was able to be completely oxidized at 295 °C, and showed no attenuation on ethane conversion ($\Delta X_{ethane} < 1\%$) during thermal cyclic tests. In addition to this, the effect of water vapor was examined. No significant change is observed during the hydrothermal cyclic tests, and the $T_{90}$ value is about 280 °C for all cycles over $MnCoO_x$-0.5 catalyst (Fig. 2g). Also, the activity almost recovered after $H_2O$ removal, which suggests the reversible deactivation of $MnCoO_x$-0.5 catalyst. This reversible deactivation can be substantiated by $C_2H_6$-$O_2$/$O_2$ + $H_2O$ TPSR results as shown in Supplementary Fig. 12. Due to its superior performance in our lab scale tests, the $MnCoO_x$-0.5 powder was chosen and mixed with $Al_2O_3$ to prepare into a suspension for monolith washcoating. A similar preparation method was also used in one of our recently published work[31]. Afterwards, a long-term stability test was performed at 350 °C (Fig. 2h). The ethane conversion slightly dropped from ca. 76 to 68% at the initial stage of the reaction either with or without water addition. After that, no deactivation was observed up to 1000 h time-on-stream (TOS) measurement, which demonstrates the superior water-resistance of monolith $MnCoO_x$-0.5 catalyst.

## Role of $MnO_2$-$Mn_xCo_{3-x}O_4$ interface

To gain a better understanding on the interfacial regions, the aberration-corrected STEM images and EELS analyses were performed to determine the structure and morphology of $MnCoO_x$ catalysts (Fig. 3, Supplementary Figs. 13–15). An enlarged image on these nanosheets yields a periodic lattice fringe of 0.48 nm, corresponding to the (111) plane of $MnCo_2O_4$, which again confirmed the successful substitution of Mn into the lattice of cubic $Co_3O_4$ (Fig. 3a). Outside the microspheres, some ultra-thin layers were noticeable with an average thickness of ca. 4–5 nm for $MnCoO_x$-0.5. The measured lattice spacing is about 0.24, 0.21, and 0.31 nm, which can be indexed to the (101), (111), and (110) planes of $MnO_2$ (PDF#24-0735), respectively (Fig. 3b). Overall, the HRTEM images provide visual evidence for the formation of $MnO_2$-$MnCo_2O_4$ interface, as illustrated in Fig. 3c, d. To better understand the chemical environment of elemental Mn and Co at

$MnO_2$-$Mn_xCo_{3-x}O_4$ interface, the EELS line-scanning was employed. The elemental distribution from electron energy loss spectra (EELS) clearly showed that Mn is evenly distributed on the shell of $MnCo_2O_4$ microsphere (Fig. 3e–g). Also, the EELS area scanning images give a direct view on the close contact between Co and Mn (Fig. 3h–k). Noted that, Mn prefers to stay on the edge of $MnCo_2O_4$ nanosheets. Next, we employed surface-sensitive technique TOF-SIMS to distinguish the chemical composition between surface and interior of $MnCoO_x$ catalyst. Supplementary Fig. 16 presents the depth profile of $^{55}Mn^+$ and $^{59}Co^+$ elements, which again confirms the enrichment of Mn on the surface of $Mn_xCo_{3-x}O_4$ microspheres. Similar conclusion was also obtained on the depth profile of $Mn^{4+}/Mn^{3+}$ and $Co^{2+}/Co^{3+}$ atomic ratio from the XPS data (Supplementary Fig. 17).

After studying the microstructure of $MnCoO_x$ catalysts, the properties of $MnO_2$-$MnCo_2O_4$ interface were explored. To attain a deeper understanding on the reactivity of $MnO_2$-$MnCo_2O_4$ interface, a platform $MnO_2$/$MnCo_2O_4$ catalyst with 1 wt% of Mn loading was synthesized. Firstly, $C_2H_6$-TPSR was carried out to study the properties and reactivity of involved O on $MnCoO_x$-0.5 (Fig. 4a). Both $CO_2$ ($m/z = 44$) and $H_2O$ ($m/z = 18$) were detected in the tested temperature range (50–500 °C). After studied the O reactivity of $MnO_2$, $MnCo_2O_4$, and $MnO_2$/$MnCo_2O_4$ references (Supplementary Fig. 18), we deduce that the evolved $CO_2$ peak below 250 °C (as indicated in the yellow box) can be ascribed to the oxygen that is located at or near $MnO_2$-$MnCo_2O_4$ interfacial region, while the high-temperature peak above 400 °C (as indicated in the pink box) is assigned to the bulk $MnCo_2O_4$ substrate. Besides, a relatively weak $CO_2$ peak appeared at 347 °C (as indicated in the blue box), suggestive of the existence of a small portion of aggregated $MnO_2$. Comparatively, the $C_2H_6$-TPSR result of $MnCoO_x$-0.5 catalyst indicates the reactive nature of surface lattice O that located at the interface of $MnO_2$-$MnCo_2O_4$. To get more insights into the activity of lattice oxygen ($O_{Latt}$) near $MnO_2$-$MnCo_2O_4$ interfacial areas, two DRIFT-MS experiments were designed. One was carried out in an $O_2$-free environment under isothermal conditions, and the other experiment was performed in transient state. Notably, $CO_2$ was detected on the $MnCoO_x$-0.5 catalyst without gas-phase $O_2$ supply, indicating the participation of lattice O at 250 °C (Supplementary Fig. 19). The transient DRIFT-MS analysis showed that the transition period for $O_2$-depletion follows the trend of $MnCoO_x$-0.5 ($\Delta t = 210$ s) > $MnO_2$/$MnCo_2O_4$ ($\Delta t = 106$ s) > $MnO_2$ ($\Delta t = 90$ s) (Supplementary Fig. 20), which is in accordance with the isotherm experiments.

Following this result $^{18}O_2$ isotopic labeling experiments were performed to monitor how the lattice oxygen was involved in ethane oxidation. The formation of $C^{16}O_2$ ($m/z = 44$) became noticeable above 65 °C, indicating the active nature of $O_{latt}$ on $MnCoO_x$-0.5 (Fig. 4b). Noted that, $C^{16}O_2$ doublet peak appeared (158 and 237 °C), which represent two types of lattice O. As the reaction proceeds, the formation of $C^{16/18}O_2$ occurs (163 °C) accompanied with the gradual decline of $C^{16}O_2$, indicating that the oxygen exchange was taking place between gas phase $^{18}O_2$ and lattice $^{16}O$ from the catalyst. Followed by this, the formation of $C^{18}O_2$ is initiated (200 °C) due to the depletion of surface lattice $^{16}O$ and the $^{18}O_2$ replenishment. The obtained isotope results emphasized the effectiveness of lattice O in $MnCoO_x$-0.5. For $MnO_2$ reference, the lattice $^{16}O$ could also participate in oxidation, but with a higher onset temperature (189 °C), an indicator of the low activity of $O_{latt}$ (Supplementary Fig. 21). Also, the presence of $C^{16}O_2$ (or $H_2^{16}O$) single peak suggested that there is only one type of lattice O participating in the reduction process, which is distinct from $MnCoO_x$-0.5. Overall, these isotopic O exchange studies suggests that the ethane oxidation is dominated by a surface Mars-van Krevelen (MVK) mechanism in both cases.

Subsequently, temporal analysis of products (TAP) was undertaken to unveil the dynamic surface change of $MnCoO_x$-0.5 and $MnO_2$ reference as a function of temperature (Fig. 4c–h, Supplementary

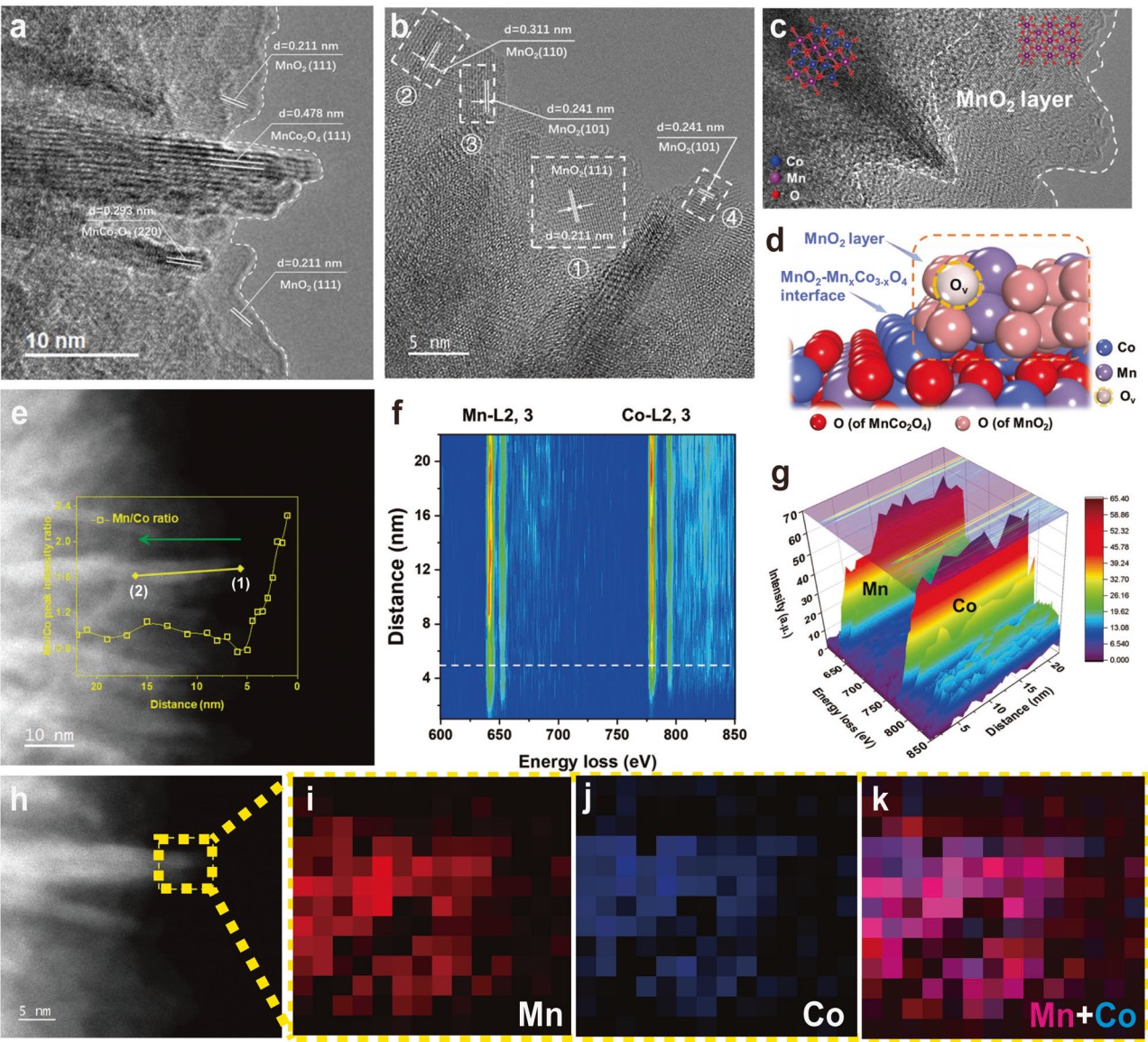

**Fig. 3 | Microstructure characterizations of MnCoO$_x$ catalyst. a, b** HRTEM images of MnCoO$_x$-0.5 catalyst at selected interfacial areas. **c** HRTEM image of MnCoO$_x$-2.0 catalyst. **d** schematic illustration of MnCoO$_x$-0.5 at interface. **e** high-angle annular dark-field (HAADF) images of MnCoO$_x$-0.5 catalyst at MnO$_2$-Mn$_x$Co$_{3-x}$O$_4$ interface with an insert showing the change of Mn/Co ratio along the yellow line from point (1) to (2), as indicated by the green arrow. **f, g** Mn-L2, 3-edge and Co-L2, 3 edge spectra as a function of line scanning distance (indicated by the green arrow on (**e**)). **h–k** EELS elemental maps of Mn, Co, and the corresponding Mn-Co overlap of MnCoO$_x$-0.5 catalyst. Scale bar, 5 nm; Red: Mn, Green: Co. (Source Data are provided as a Source Data file).

Fig. 22). During each test, a small quantity of reactant mixture (5 ml, $C_2H_6$ + $^{18}O_2$ + He) was injected in the temperature range of 200–400 °C to facilitate the scrambling of $^{18}O/^{16}O$ atoms, thereby making it possible to capture the initial catalytic behavior of the material. Despite the quantitative difference in product distribution between steady-state and TAP experiments, the general selectivity trends were consistent. Note that the amounts of $^{16}O$-containing products ($C^{16}O_2$ and $C^{16/18}O_2$, accounts for >95%) significantly exceed that of $C^{18}O_2$ at 200 °C on the MnCoO$_x$-0.5 catalyst. Upon combining with the results obtained from $C_2H_6$-TPSR analysis, we can confidently verify that the majority of the participated O arises from the lattice O that resides at MnO$_2$-MnCo$_2$O$_4$ interface, exhibiting a remarkable reactivity in promoting oxidation reactions, particularly at relatively lower temperatures. Also, we found that the activity of lattice O on MnCoO$_x$-0.5 is significantly higher than that of bulk MnO$_2$ (insert of Fig. 4d). At 250 °C, the surface lattice $^{16}O$ is quickly consumed as indicated by the increase of $C^{16/18}O_2$ and $C^{18}O_2$. However, once the temperature is above 300 °C, the amount of $^{16}CO_2$

slightly increased due to the enhanced bulk phase O migration/diffusion to refill the surface O$_v$ at high temperature. Thereby, we can infer that the replenishment of O$_v$ originates from a conjugated effect both from the gaseous O$_2$ and bulk phase O migration/diffusion, in which the contribution from the latter could be enhanced at high temperature. Also, the results evidently conclude that the lattice O stayed at MnO$_2$-MnCo$_2$O$_4$ interfaces plays a crucial role for low-temperature ethane activation.

Aside from this, the in-situ XPS analyses (Fig. 5a, Supplementary Table S11) showed that the ratio of $Co^{2+}/Co^{3+}$ quickly increased from 0.55 (fresh sample at RT) to 0.64 ($C_2H_6$ at 200 °C) with no more change above 200 °C, perhaps due to the efficient electron transfer from the absorbed $C_2H_6$ to the positively charged Co ions. While from Mn 2$p$ spectra, we observed the significant increase of $Mn^{\delta+}/Mn^{4+}$ ratio from 0.94 (fresh sample at RT) to 3.18 ($C_2H_6$ at 400 °C) accompanied by the shifting of $Mn^{\delta+}$ peak towards lower B.E., suggesting the consumption of lattice O on MnO$_2$ domains during H abstraction, thereby resulting

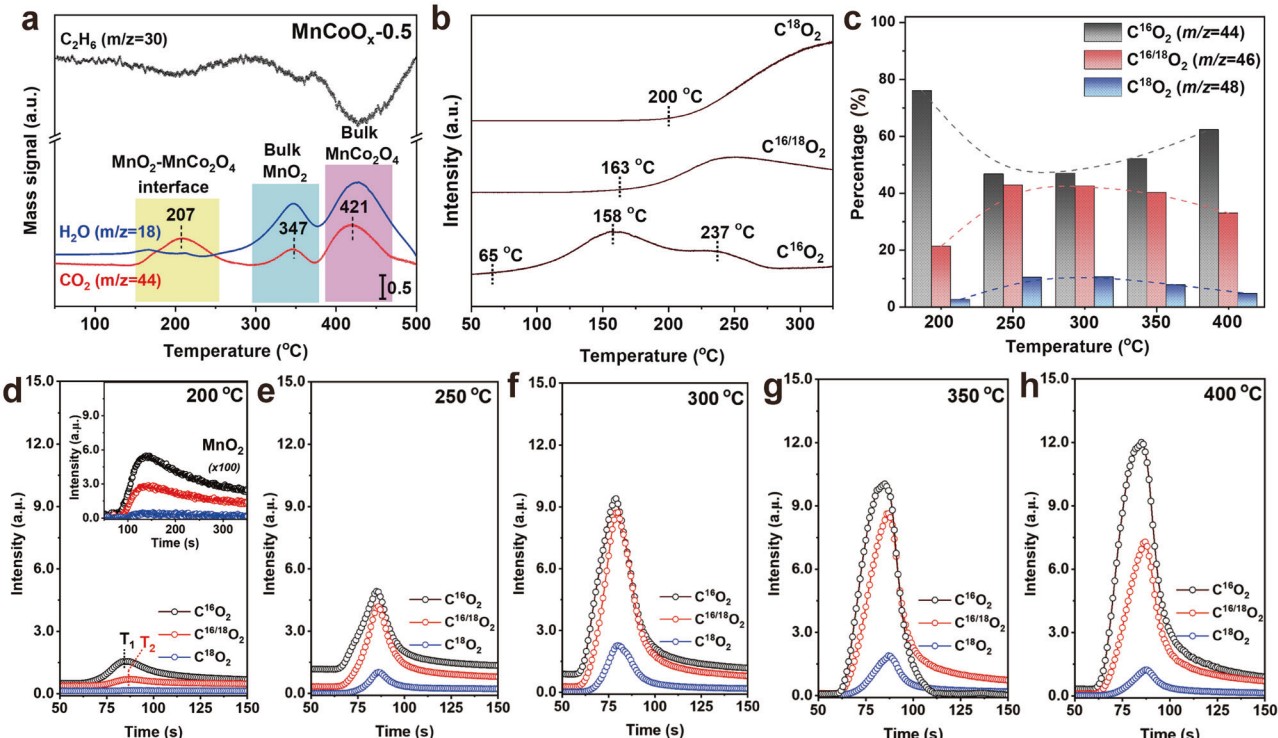

**Fig. 4 | Role of MnO₂-MnCo₂O₄ interface for ethane oxidation. a** $C_2H_6$-TPSR-MS profile. **b** $^{18}O$ isotopic labeling experiment in the temperature programmed oxidation of ethane over MnCoO$_x$-0.5 catalyst. **c–h** Temporal analysis of products (TAP) of ethane oxidation over MnCoO$_x$-0.5 as a function of temperature from 200 to 400 °C (the insert of **d** represents the TAP analysis of MnO₂ reference at 200 °C, $T_1$ stands for the maximum temperature of generated $C^{16}O_2$, $T_2$ stands for the maximum temperature of produced $C^{16/18}O_2$). (Source Data are provided as a Source Data file).

in a coordination change on Mn species. Once $C_2H_6/O_2$ mixture was introduced into the system, the $Mn^{\delta+}/Mn^{4+}$ ratio slightly increased from 3.18 ($C_2H_6$ at 400 °C) to 2.11 ($C_2H_6/O_2$ at 400 °C), while the $Co^{2+}/Co^{3+}$ ratio almost went back to its original states. This result again indicates the participation of lattice oxygen from MnO₂ layer. Also, the in-situ XPS results revealed that the $O_\alpha$ (lattice O) peak gradually shifts towards lower B.E. with increased $C_2H_6$ reduction temperature, indicating the weakened interaction between Co/Mn and O atoms, potentially resulting in an increase in oxygen vacancies[48]. The $O_\alpha$ species gradually consumed during $C_2H_6$ reduction from 85.4% (fresh catalyst at RT) to 74.8% ($C_2H_6$ at 400 °C). This observation further confirmed the participation of lattice O species over the MnCoO$_x$-0.5 catalyst during $C_2H_6$ oxidation, which is consistent with the isotopic labeling experiments.

Next, the adsorption of $C_2H_6$ over MnCoO$_x$ catalysts was investigated. As shown in the time-resolved DRIFT spectra (Fig. 5b), the intensity of ethane adsorption bands (3000 cm$^{-1}$) gradually increased with time-on-stream operation to reach a steady-state level. The MnCoO$_x$-0.5 exhibited the strongest ethane adsorption capacity compared to MnCo₂O₄ and MnO₂ references (Supplementary Fig. 23). Interestingly, the time it took to detect ethane follows the order of MnCoO$_x$-0.5 (11.0 min) > MnCo₂O₄ (7.5 min) > MnO₂ (4.0 min), indicating that more $C_2H_6$ are adsorbed/activated over MnCoO$_x$-0.5 catalyst. Again, CO₂ was detected at 2300–2400 cm$^{-1}$, which indicates the participation of lattice O. Moreover, $C_2H_6$-TPD was employed to address the chemisorption behavior of $C_2H_6$ over MnCoO$_x$. As shown in Supplementary Fig. 24, CO, CO₂, and H₂O as main products were detected due to the reduction of $C_2H_6$ from lattice O, but with different desorption temperatures. Also, the integrated peak area of produced C-related species followed a decreasing trend of MnCoO$_x$-0.5 > MnCo₂O₄ > MnO₂, suggesting that more ethane was preserved over MnCoO$_x$-0.5 catalyst.

Furthermore, the ethane oxidation activity of MnCoO$_x$-0.5 is compared to MnO₂/MnCo₂O₄, MnCo₂O₄, and MnO₂ references, to identify the catalytic contribution of MnO₂-MnCo₂O₄ interface (Supplementary Fig. 25). Clearly, the areal rate of MnCoO$_x$-0.5 (1.35 × 10$^{-2}$ µmol m$^{-2}$ s$^{-1}$, 220 °C) is close to that of MnO₂/MnCo₂O₄ (1.14 × 10$^{-2}$ µmol m$^{-2}$ s$^{-1}$, 220 °C), indicating that the high conversion of MnCoO$_x$-0.5 catalyst may result from the presence of MnO₂-MnCo₂O₄ interface. Noted that the temperature of T50 dramatically reduced to 304 C for the physically mixed MnO₂ and MnCo₂O₄ (referred to as Phy-MnCo₂O₄-MnO₂) catalyst compared to pure MnO₂. A similar performance was obtained on the layer-packed MnCo₂O₄-MnO₂ (refers to as LP_MnCo₂O₄-MnO₂, $T_{50}$ = 311 °C). However, the catalytic activity of MnCo₂O₄ and MnO₂ mixtures was lower than that of the MnO₂/MnCo₂O₄ model catalyst regardless of their mixing methods, indicating the significant role of interfacial sites due to the proximity between the two components. In this regard, it is imperative to study the correlation of MnO₂-MnCo₂O₄ interface with catalytic properties. Therefore, several control experiments were designed by annealing the MnCoO$_x$ precipitates under N₂ and air, respectively. It was found that the number of MnO₂-MnCo₂O₄ interfacial sites can be altered based on the strong O₂ affinity of Mn, which is similar to the synthesis of core/shell Au/MnO and PtFe-FeO$_x$/TiO₂ catalysts[25,49]. From XPS analysis, we know that there are more high valence Mn species appeared on the surface of the air calcined MnCoO$_x$-0.2 catalyst compared to the N₂-treated one, as evidenced by the high AOS value and Mn$^{4+}$/Mn$^{3+}$ ratio (Supplementary Fig. 26). Hence, it is reasonable to deduce that more Mn species diffuse out onto the Mn$_x$Co$_{3-x}$O₄ spinel surface forming MnO₂ domains due to the strong O₂ driving force and consequently, creating more MnO₂-MnCo₂O₄ interfaces. Also, the $C_2H_6$-TPD results showed that the MnCoO$_x$-0.2-Air has a strong $C_2H_6$ storage capacity compared to that of MnCoO$_x$-0.2-N₂ (Supplementary Fig. 27). Eventually, a positive correlation was established between ethane

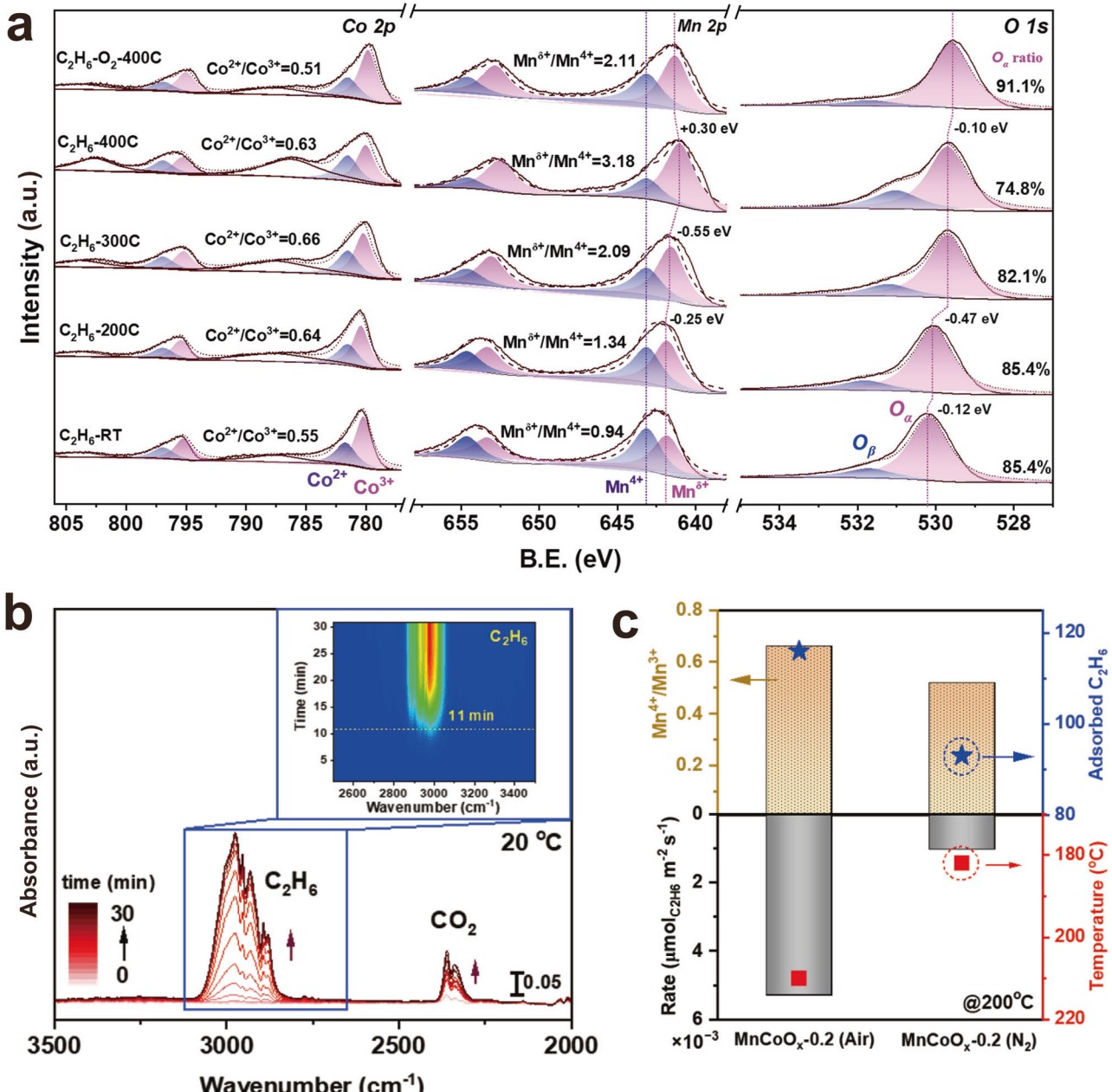

**Fig. 5 | Properties of MnO₂-MnCo₂O₄ interface for ethane oxidation. a** in-situ XPS analysis of MnCoOₓ-0.5 catalyst under different gas atmospheres. **b** DRIFT spectrum of ethane adsorption over MnCoOₓ-0.5. **c** The correlation between ethane conversion rate (or temperature at constant rate of 0.21 μmol $g_{cat}^{-1}$ $s^{-1}$) and Mn⁴⁺/Mn³⁺ ratio (or C₂H₆ adsorption capacity). (Source Data are provided as a Source Data file).

conversion rate and Mn⁴⁺/Mn³⁺ ratio, which proved the highly effective of MnO₂-MnCo₂O₄ interface in catalyzing ethane oxidation (Fig. 5c, Supplementary Fig. 28).

## Surface mechanism

Density functional theory (DFT) calculations were carried out to further assess the effects of MnO₂-MnCo₂O₄ interface and provide information on how the constructed interface contributes to the catalytic behaviors, especially in terms of the interactions with reactants. Similar to one of our recent work[31], we constructed the MnCo₂O₄ crystal structure by replacing part of the octahedral Co atoms of cubic Co₃O₄ with Mn. As shown in supplementary Fig. 29a, the Type (II) model was found to be the most stable structure in our calclation by substituting octahedral Co³⁺ with Mn³⁺, as demonstrated by the lowest relative energy per Mn atom in the proposed MnCo₂O₄ models. The

obtained lattice parameter of MnCo₂O₄ spinel is enlarged from 8.07 to 8.14 Å, which is consistent with the XRD results. Meanwhile, the bulk MnO₂ models exposed with (111), (110), and (101) facets as well as the MnCo₂O₄ (111) facets (Supplementary Fig. 29b, c) were built to correlate with what we observed from the HRTEM images (Fig. 3). After analyzing the termination stability of MnO₂ and MnCo₂O₄, the optimized interfacial models of MnO₂-MnCo₂O₄ were established by taking MnCo₂O₄-111-A as the underlying substrate and intercepting a structural unit from MnO₂-111-C, MnO₂-110-B, and MnO₂-101-B as the upper cluster (named as MnCo₂O₄/MnO₂-111-C, MnCo₂O₄/MnO₂-110-B, and MnCo₂O₄/MnO₂-101-B, respectively, see details in Supplementary Figs. 30–31). Figure 6a showed the adsorption energy of C₂H₆ and O₂ as well as the oxygen vacancy formation energy ($E_{Ov}$) on the bulk MnO₂ and MnCo₂O₄/MnO₂ catalyst models. Taken MnCo₂O₄/MnO₂-111-C as an example, we can clearly see that the adsorption energy of C₂H₆ at

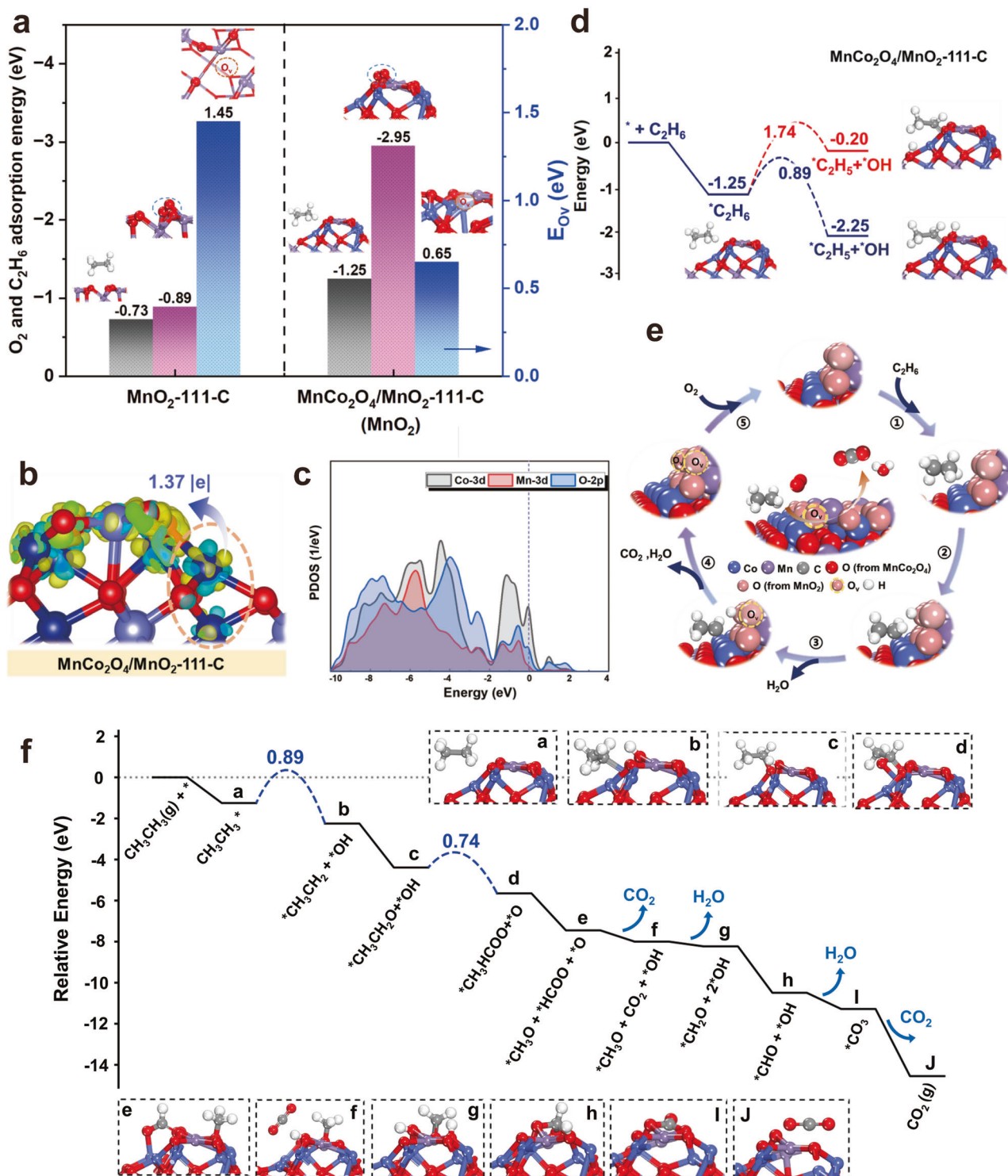

**Fig. 6 | Mechanistic study of ethane oxidation over the MnCoO$_x$-0.5 catalyst.** **a** The calculated adsorption energy of C$_2$H$_6$, O$_2$, and the O$_v$ formation energy on the bulk MnO$_2$−111-C and MnCo$_2$O$_4$/MnO$_2$-111-C catalyst models (Note: The adsorption energy of C$_2$H$_6$ was obtained by adsorbing C$_2$H$_6$ at the interfacial region of MnCo$_2$O$_4$/MnO$_2$-111-C model; the O$_2$ adsorption energy was obtained by adsorbing O$_2$ on the upper MnO$_2$ cluster of MnCo$_2$O$_4$/MnO$_2$-111-C model). **b** the calculated differential charge density between O atom in the upper MnO$_2$ cluster and the interfacial Co atom of MnCo$_2$O$_4$/MnO$_2$-111-C. **c** the calculated projected density of states (PDOSs) of Co-3$d$, Mn-3$d$ and O-2$p$ orbital on the MnCo$_2$O$_4$/MnO$_2$-111-C (the Fermi level was set to zero and the isosurface value was set to 0.005 e Å$^{-3}$; the cyan and yellow regions represent positive and negative charges, respectively). **d** energy

profiles for the dissociation of the first C-H bond of C$_2$H$_6$ over MnCo$_2$O$_4$/MnO$_2$-111-C model (red line: H abstraction of adsorbed C$_2$H$_6$ at the interfacial region of MnCo$_2$O$_4$/MnO$_2$ model catalyst; blue line: H abstraction of adsorbed C$_2$H$_6$ at the upper MnO$_2$ cluster of the MnCo$_2$O$_4$/MnO$_2$ model catalyst). **e** schematic illustration of the reaction mechanism of ethane oxidation over MnCoO$_x$-0.5 catalyst (① C$_2$H$_6$ adsorption; ② Initiated 1$^{st}$ H abstraction; ③ Continuous H abstraction; ④ CO$_2$ and H$_2$O desorption; ⑤ Refilling O$_v$ by O$_2$). **f** Energy diagram of the optimal reaction paths for ethane oxidation on MnCo$_2$O$_4$/MnO$_2$-111-C catalyst surface and the optimized structures of all species involved. (Source Data are provided as a Source Data file).

the interface of $MnCo_2O_4/MnO_2$-111-C model (-1.25 eV) is negatively higher than the corresponding bulk $MnO_2$-111-C (-0.73 eV), indicating the preferential adsorption of $C_2H_6$ on the former catalyst. Also, the adsorption of $O_2$ at the interface of $MnCo_2O_4/MnO_2$-111-C model (−1.01 eV) is negatively less than that of $C_2H_6$ (-1.25 eV), indicating that $O_2$ cannot compete with $C_2H_6$ for the adsorption at $MnO_2$-$MnCo_2O_4$ interface (Supplementary Fig. 32a). Similar results were also obtained on other interfacial models ($MnCo_2O_4/MnO_2$-110-B and $MnCo_2O_4/MnO_2$-101-B, Supplementary Fig. 32b, c).

Interestingly, we found that the $O_2$ molecule is prone to be activated on the topmost $MnO_2$ domain of the $MnCo_2O_4/MnO_2$-111-C catalyst as evidenced by the partial electron transfer from $MnCo_2O_4$ sublayer to $MnO_2$ via the interfacial Co cations to O anions that located at the adjacent of $MnO_2$ cluster (1.37 |e|), as shown in Fig. 6b. In addition, the calculated projected density of states (PDOS) shows a upshift of O $p$-band near Fermi level, indicating a strong interaction between Co 3d and O 2p orbitals. The enhanced $C_2H_6$ adsorption can also be explained by the strong hybridization between O 2$p$ and Co-3$d$/Mn-3$d$ orbitals (Fig. 6c). To further confirm this, we carried out a crystal orbital Hamilton population (COHP) calculation to get a quantitative analysis of the interfacial O-Co bond interaction of $MnCo_2O_4/MnO_2$ interfacial models (Supplementary Fig. 33). The integral values below the Fermi level are -1.74 over $MnCo_2O_4/MnO_2$-111-C catalyst, which again demonstrated the significant hybridization between O and Co sites. Additionally, we calculated the adsorption energy of $C_2H_6$ on the $x$Co/$MnO_2$ ($x = 1$–$2$) model catalysts by varying the Co content on different $MnO_2$ planes to gain a better understanding of the Co-O-Mn sites and their effects on $C_2H_6$ activation (Supplementary Fig. 34). Noticeably, the adsorption strength of $C_2H_6$ increases with increasing the Co substitution contents, indicating a positive effect of Co sites on $C_2H_6$ adsorption, which aligns with the experimental results (Supplementary Fig. 35). Meanwhile, the lowest oxygen vacancy formation energy ($E_{Ov} = 0.65$ eV) was obtained on the uppermost $MnO_2$ domain of $MnCo_2O_4/MnO_2$-111-C model compared to other proposed catalyst models, which confers a better $O_2$ adsorption ability on this catalyst (Supplementary Fig. 32d). Also, the average Mn-O bond length of $MnCo_2O_4/MnO_2$-111-C (1.94 Å) is larger than that of $MnO_2$-111 (1.83 Å), which implied a high O mobility on the former model (Supplementary Fig. 36). Therefore, the significant influence from the underlying spinel $MnCo_2O_4$ was identified.

To understand the underlying mechanism of $C_2H_6$ oxidation over the $MnCoO_x$-0.5 catalyst, a detailed discussion of the first C-H bond dissociation of $C_2H_6$ was carried out on the $MnCo_2O_4/MnO_2$-111-C model, because this step was typically being regarded as the kinetically relevant step[31]. As shown in Fig. 6d, two reaction pathways were proposed based on the position of the abstracted H, either bind to the O sites of the upper $MnO_2$ cluster or to the underlying $MnCo_2O_4$ substrate, eventually forming OH groups. The obtained results showed that the energy barrier ($\Delta E_{TS}$) of C-H bond cleavage on the O sites of $MnO_2$ cluster ($\Delta E_{TS}$: 0.89 eV) is lower than that on the $MnCo_2O_4$ substrate ($\Delta E_{TS}$: 1.74 eV), indicating that the former route is kinetically more favorable. Moreover, the formation of OH group from $C_2H_6$ dissociation on the upper $MnO_2$ clusters is thermodynamically more favorable by releasing energy of 1.00 eV, whereas the OH group formation on the $MnCo_2O_4$ substrate is endothermic by 1.05 eV. Therefore, the lattice oxygen species of $MnO_2$ domain plays a significant role in $C_2H_6$ oxidation, as evidenced by both experimental and DFT results. Similar trends were also obtained on the other two interfacial models ($MnCo_2O_4/MnO_2$-110-B and $MnCo_2O_4/MnO_2$-101-B, Supplementary Fig. 37). Compared to the $MnCo_2O_4$-111-A model without $MnO_2$ domain, the C-H bond dissociation barrier (1.27 eV) is higher than that obtained on the $MnCo_2O_4/MnO_2$-111-C interfacial model, inferring an interfacial engineering of $MnO_2$-$Mn_xCo_{3-x}O_4$ catalyst to boost ethane oxidation. Here, a schematic illustration of the reaction mechanism was proposed and illustrated in Fig. 6e. Subsequently, the energy

diagram of elementary steps for ethane oxidation along the reaction pathways was calculated to gain a deeper understanding on the $MnO_2$-$MnCo_2O_4$ interfacial system, as illustrated in Fig. 6f. After dissociating the first C-H bond of $C_2H_6$, the generated $^*CH_3CH_2$ species is prone to bond on Co sites that located at the interface of $MnO_2$ and $MnCo_2O_4$ substrate (Fig. 6f, b), which aligns with the $C_2H_6$-TPSR results. Then, the adsorbed $^*CH_3CH_2$ changes its adsorption site from interfacial Co to the lattice $O^*$ of upper $MnO_2$ cluster to form $^*CH_3CH_2O$ (Fig. 6f, c), which is proved to be thermodynamically favorable by releasing an energy of 2.14 eV. This calculation is in line with our in-situ XPS results, which implies that further dehydrogenation mostly occurs on the upper $MnO_2$ domains. After that, the produced $^*CH_3CH_2O$ entities undergo further dehydrogenation, resulting in the formation of $^*CH_3HCOO$ intermediates (Fig. 6f, d). These intermediates subsequently decompose into $^*CH_3O$ and $^*HCOO$ by breaking the C-C bonds, releasing an energy of 1.8 eV. Finally, the continuous dehydrogenation of $^*CH_3O$ and $^*HCOO$ leads to the formation of $^*CH_2O$,$^*CHO$, $CO_2$, and $H_2O$ species, showing a downhill energy profile. Overall, DFT results are consistent with the in-situ DRIFT studies (Supplementary Fig. 39) and confirm that the first C-H bond cleavage of $C_2H_6$ is the rate-determining step in ethane combustion on the $MnCo_2O_4/MnO_2$ interfacial catalyst, which has a barrier of 0.89 eV. Based on the above analyses, we can reasonably conclude that the simultaneous enhancement on ethane adsorption/activation and lattice O mobility of $MnCoO_x$-0.5 catalyst is proved to be the main reason of achieving an excellent activity in ethane oxidation, which is ingeniously controlled by interfacial engineering.

## Discussion

In summary, we have successfully developed the $MnCoO_x$ catalyst by a facile chemical reduction synthesis method, which shows the highest specific reaction rate in ethane combustion beyond all the reported non-noble metal catalysts, as well as an excellent long-term stability up to 1000 h even under humid conditions. Mn with strong O affinity tends to diffuse out onto the spinel surface forming $MnO_2$ domains during an $O_2$-rich environment. The established interaction between $MnO_2$ and $Mn_xCo_{3-x}O_4$ triggers the construction of interfacial sites. Surprisingly, the Co sites on the established hierarchical interface of $MnO_2$-$Mn_xCo_{3-x}O_4$ exhibit a preferential adsorption on ethane; while, the $MnO_2$ layer displays a strong ability of doing H abstraction on their active lattice O, and further proceed the ethane oxidation through a redox pathway at interfacial regions. Revealing the essential role of interface provides an effective strategy of regulating the coordination environment of involved components as well as their electron transfer ability.

## Methods

### Materials

Potassium permanganate (VII) ($KMnO_4$ powder, ≥99.0 %), manganese (II) nitrate tetrahydrate ($Mn(NO_3)_2 \cdot 4H_2O$, ≥97%) and cobalt (II) nitrate hexahydrate ($Co(NO_3)_2 \cdot 6H_2O$, ≥97%) purchased from Sinopharm, were used as received without further purification.

### Catalyst preparation. (1) Synthesis of $MnCoO_x$

(1) **Synthesis of $MnCoO_x$:** $MnCoO_x$ were synthesized by a redox-controlled synthesis method ($Mn^{7+} + 3Co^{2+} \rightarrow Mn^{4+} + 3Co^{3+}$). In a typical synthesis process, the Mn (VII) solution was prepared by dissolving certain amounts of $KMnO_4$ into 1000 mL deionized water under magnetic stirring for 30 min at 70 °C. The Co (II) solution was prepared by dissolving specific amounts of $Co(NO_3)_2 \cdot 6H_2O$ into aqueous solution with certain amounts of potassium citrate under magnetic stirring. Subsequently, the prepared Co precursor solution was added dropwise into the $KMnO_4$ solution at a specific injection speed to control the reduction process. After completed the injection, the mixed solution was keeping stirring for another 2 h at 80 °C. Then, the mixed

solution was maintained under ambient conditions. After aging for a few hours, the black precipitate was collected by filtration, and washed by deionized water and absolute ethanol three times before drying. After that, the precursor was subjected to an annealing treatment in static air at 350 °C for 2 h at a ramping rate of 1 °C min⁻¹. Finally, the resulting catalysts were washed by 1M $NH_4NO_3$ solution for 2 h at room temperature under stirring to remove K ions prior to the catalytic tests. The obtained catalysts were denoted as $MnCoO_x$-z, where z represents the nominal molar ratio of Mn/Co. The synthesis parameters of $MnCoO_x$ catalysts are given in Table S10. (2) **Synthesis of $Co_3O_4$**: A typical precipitation method was employed to prepare $Co_3O_4$ reference by adding ammonia (1 mol L⁻¹) into cobalt nitrate solution. After vigorous stirring for 2 h, the solution was filtered and washed three times by deionized water and ethanol. The obtained precipitate was dried at 70 °C for 12 h and followed by annealing in air at 400 °C for 3 h. (3) **Synthesis of $MnO_2$**: $MnO_2$ nanoparticles (NPs) were synthesized by using $KMnO_4$ solution as Mn precursor to take K effects into account. The precursor was prepared according to the procedure described elsewhere[50,51]. Typically, oleic acid (10.0 mL) was added to $KMnO_4$ solution (0.0126 mol L⁻¹). After vigorous stirring for 30 min, the emulsion was washed by water and ethanol three times to remove residuals. Then, the product was dried in air at 80 °C overnight before calcinated in air. Finally, the obtained precursor was treated at 200 °C in air for 5 h. (4) **Synthesis of $MnCo_2O_4$ and $MnO_2/MnCo_2O_4$**: $MnCo_2O_4$ support was synthesized by a conventional precipitation method. Typically, ammonia (1 mol L⁻¹) was added dropwise to the solution of $Mn(NO_3)_2 \cdot 4H_2O$ (0.19 mol L⁻¹) and $Co(NO_3)_2 \cdot 6H_2O$ (0.38 mol L⁻¹). After vigorous stirring for 6 h, the mixture was filtered and dried. The obtained powder was calcined at 350 °C for 4 h. The resulting sample was denoted as $MnCo_2O_4$. A supported 1%$MnO_2$/$MnCo_2O_4$ catalyst was prepared by chemically reducing Mn on the obtained $MnCo_2O_4$ support. Firstly, 1 g of $MnCo_2O_4$ support was dispersed in 25 mL deionized water. After that, 7.286×10⁻⁵ mol $KMnO_4$ was added into the $MnCo_2O_4$ dispersed solution and stirring for 30 min. Followed by this, 1.09×10⁻⁴ mol $Mn(NO_3)_2$ was added to reduce $KMnO_4$. The mixture was stirred for another 30 min before increasing the temperature to 70 °C for 2 h. The obtained precipitate was filtered and washed by water and ethanol three times. The obtained precursor was firstly dried at room temperature for 24 h, and then dried at 70 °C for another 12 h. Eventually, the as-prepared sample was calcined at 350 °C for 2 h.

## Characterization

The specific surface area, pore volume, and averaged pore size were determined from $N_2$ adsorption-desorption isotherms measured at -196 °C using a Micromeritics ASAP 2020 analyzer. All samples were degassed at 200 °C (100 μm Hg) for 6 h. The specific surface area ($S_{BET}$) was calculated from the measured $N_2$ isotherm using the Brunauer-Emmett-Teller (BET) equation applied in a relative pressure range (P/$P_o$) of 0.01-0.35. The total pore volume ($V_{total}$) was obtained from $N_2$ uptake at a relative pressure of P/$P_o$ = 0.99. The averaged pore size was calculated by $4V_{total}/S_{BET}$. The elemental analysis was conducted by inductively coupled plasma-optical emission spectroscopy (ICP-OES). An acid digestion was conducted by aqua regia at 100 °C to dissolve all metals.

Powder X-ray diffraction (XRD) patterns were performed on a Rigaku SmartLab 9 kW diffractometer with Cu Kα ($\lambda$ = 1.5406 Å) radiation operating at 45 kV and 200 mA to determine the bulk structure of the synthesized materials. The scanning speed was set at 10 s/step with 2$\theta$ in the range of 10° to 70°.

Raman scattering spectra were collected on a DRX Microscope instrument (Thermo Fisher Scientific) with an exciting wavelength of $\lambda_{ex.}$ = 532 nm equipped with a charge coupled device (CCD) detector at ambient conditions. The scanning range was set at 100–1000 cm⁻¹ with resolution of 1.0 cm⁻¹. The attenuated total reflection Fourier

Transform Infrared Spectroscopy (ATR FT-IR) spectrum were collected by Thermo Nicolet iS50 spectrometer from 400 to 4000 cm⁻¹. The electron paramagnetic resonance (EPR) spectra were obtained on Bruker EPR equipment (model a220-9.5/12) at room temperature by detecting the unpaired electron.

Field emission scanning electron microscopy (FESEM) images were obtained with a Hitachi SU8220 instrument with an acceleration voltage of 5 kV to reveal the morphology of $MnCoO_x$ catalysts. The average particle size was calculated by accounting >100 particles/clusters and then fitting the measured size to a normal distribution. Energy dispersive X-ray (EDX) elemental analyses were used to reveal the elemental composition of $MnCoO_x$ catalysts. The high-resolution transmission electron microscopy (HRTEM) images and electron energy loss spectroscopy (EELS) spectrum images were recorded on Thermis ETEM (thermos Scientific) operated at 300 kV in dual EELS mode with energy resolution of 1.3 eV. A time flight secondary ion mass spectrometer (TOF-SIMS; TESCAN Amber) was used in a dynamic mode to get a depth profile of Mn and Co elements in $MnCoO_x$ sample.

X-ray photoelectron spectroscopy (XPS) was used to analysis the relative abundance and chemical state of the surface components of $MnCoO_x$ catalysts (Thermo Fisher ESCALAB™ xi⁺). The monochromatic Al Kα was used as the photo source (1486 eV). Binding energies were corrected for surface charging by referring to C 1s peak at 284.8 eV. For depth profile analysis, Ar⁺ sputtering was performed with an acceleration voltage of 500 eV with an irradiation area of 2 mm×2 mm. Also, the in-situ XPS analysis was conducted to investigate the evolution of $MnCoO_x$ catalyst during ethane oxidation. XPSPEAK41 software was used to conduct peak deconvolution. The experimental peaks were decomposed though mixing Gaussian-Lorentzian functions (80%-20%) after Shirley background subtraction. The relative ratio of each element with different valence states was calculated based on the peak areas.

Both hydrogen temperature-programmed reduction ($H_2$-TPR) and oxygen temperature-programmed desorption ($O_2$-TPD) were performed on a Micromeritics AutoChem II 2920 analyzer equipped with a TCD detector. For $H_2$-TPR measurement, about 100 mg sample was loaded and pretreated in He at 200 °C for 2 h. After cooling down, the analysis was conducted under 10%$H_2$/Ar flow from 50 to 700 °C with a ramping rate of 10 °C min⁻¹. Similar to $H_2$-TPR tests, $O_2$-TPD was conducted by firstly purging with He flow at 100 °C for 30 min to remove moisture and subsequently, switch to 10%$O_2$/He for 1 h. After that, the temperature was cooling down to room temperature in He flow to remove the physiosorbed $O_2$ and stabilize the detector baseline. Eventually, the temperature was programmed from 50 to 800 °C at a ramping rate of 10 °C min⁻¹ in He flow. Ethane temperature-programmed surface reduction ($C_2H_6$-TPSR) was carried out in a fixed-bed reactor with a mass spectrometer (MKS-Cirrus3, USA) to investigate the property of $MnO_2/MnCo_2O_4$ interface. The catalyst was firstly pretreated by 10%$O_2$/Ar flow (40 mL min⁻¹) at 300 °C for 1 h. After cooling down, the inlet gas was switched to 10 vol%$C_2H_6$/He (50 mL min⁻¹) for 30 min to stabilize the baseline. After that, a temperature-programmed reduction was conducted from 50 to 400 °C at a ramping rate of 10 °C min⁻¹. The MS signals of $C_2H_6$ (m/z = 30) and $CO_2$ (m/z = 44) were recorded accordingly. Ethane temperature-programmed desorption ($C_2H_6$-TPD) were performed on the same instrument. Typically, about 0.1 g of catalyst was pretreated under 10 vol%$O_2$/Ar (40 mL min⁻¹) flow at 300 °C for 1 h to remove the surface adsorbed water. After cooling down to 50 °C, the sample was exposed to 10 vol%$C_2H_6$ for 30 min. Next, the inlet gas was switched to He flushed for another 30 min. After being saturated with $C_2H_6$, the temperature was subsequently ramped from 50 to 650 °C at a rate of 10 °C min⁻¹ in He flow (50 ml min⁻¹). The desorbed $C_2H_6$ (m/z = 30), $CO_2$ (m/z = 44), CO (m/z = 28), and $H_2O$ (m/z = 18) were monitored by the mass spectrometer. For $C_2H_6$-$O_2$-TPSR and $C_2H_6$-$O_2$ + $H_2O$-TPSR experiments, the pretreated catalysts (ca. 0.1 g) were flushed with

$10\%C_2H_6$/He (50 mL min$^{-1}$) flow at 50 °C for 1 h, followed by He (50 mL min$^{-1}$) purging for 30 min. After that, the catalyst was heated from 50 to 400 °C under $O_2$ flow with and without $H_2O$ addition (10 vol % $O_2/N_2$, 5 vol% $H_2O$, total flow rate 40 mL min$^{-1}$).

CO chemisorption was used to determine the number of active sites (#CO, μmol g$^{-1}$) on a Quantachrome ChemBETPulsar analyzer. Samples were purged with He at 100 °C for 30 min to remove moisture, and then reduced by $5\%H_2$/Ar flow at 250 °C for 1 h. Afterwards, CO titration was performed by using thermal conductivity detector (TCD) as detector. The turnover frequency (TOF) defined as the number of alkane ($CH_4$, $C_2H_6$, and $C_3H_8$) molecules converted per active site per second, was calculated based on the following equation.

$$TOF = \frac{F_{CnH2n+2,inlet} \cdot X_{CnH2n+2}}{\#CO \cdot M} \quad (1)$$

where $F_{C(n)H(2n+2), inlet}$ represented the inlet flow rate (mol s$^{-1}$) of $C_nH_{2n+2}$, $M$ represents the molecular weight (g mol$^{-1}$) of alkane.

In-situ diffuse reflectance infrared Fourier transform spectroscopy (in-situ DRIFTs) experiments were performed on a Thermo Nicolet iS50 spectrometer equipped with mercury cadmium telluride (MCT) detector. Prior to each experiment, the catalysts were pretreated at 300 °C in $10\%O_2/N_2$ (50 ml min$^{-1}$) flow for 30 min and quickly cooling down to room temperature. After that, the experiment was performed under $1\%C_2H_6/10\%O_2/89\%N_2$ mixture to observe the evolution of reactants/products and intermediates. All the spectra were collected at a resolution of 16 cm$^{-1}$ with 32 scans in the temperature range of 50–350 °C. In addition, we examined the adsorption behavior of ethane over $MnCoO_x$-0.5 catalyst and references at 25 °C using the same setup. To investigate the interfacial property of $MnCoO_x$ catalyst, DRIFT was coupled with MS (DRIFT-MS) to perform ethane oxidation at isotherm conditions without $O_2$ feed ($1\%C_2H_6$ balanced by He, 250 °C). The MS signals of products were collected as a function of time.

Steady-state isotopic labeling experiments were performed in a fixed-bed reactor. 0.1 g of catalyst was pretreated in air at 300 °C for 1 h with a gas flow rate of 50 mL min$^{-1}$ and then flushed by $N_2$ for 30 min to clean the adsorbed $O_2$. After cooling down to 50 °C, the mixed gas of 1 vol%$^{18}O_2$ and $C_2H_6$ (1 ml min$^{-1}$) balanced by He was introduced with a total gas flow rate of 50 ml min$^{-1}$. After the baseline of MS signal was stabilized for 15-20 min, the reactor was heated from 50 to 350 °C at a ramping rate of 10 °C min$^{-1}$. During this process, the produced oxygen containing products ($C^{18}O_2$ ($m/z = 48$), $C^{16/18}O_2$ ($m/z = 46$), $C^{16}O_2$ ($m/z = 44$), $H_2^{18}O$ ($m/z = 20$), $H_2^{16}O$ ($m/z = 18$)) were monitored online by mass spectrometer. Transient mechanistic studies: Ethane oxidation was investigated in the temporal analysis of products (TAP) in pulse mode over $MnCoO_x$ and bulk $MnO_2$. Similar to steady-state isotopic labeling experiments, an oxidation treatment was conducted prior to the pulse experiments$^{18}O_2$:$C_2H_6$ = 1:1 mixture was pulsed in the temperature range of 200–400 °C with a stepwise of 50 °C for $MnCoO_x$ and 100 °C for $MnO_2$.

## Catalytic reaction

Low-chain alkane combustion was carried out in a continuous flow packed bed reactor ($\Phi = 8$ mm) to assess the catalytic activities of the $MnCoO_x$ catalysts. 200 mg catalyst (40–60 mesh) was used for each activity test. The temperature was controlled by a K-type thermocouple. The reactant gas contains 3000 ppm $C_2H_6$ ($CH_4$ or $C_3H_8$) balanced by air and $N_2$ ($O_2:N_2 = 11:89$) at a flow rate of 200 ml min$^{-1}$ (weight hourly space velocity (WHSV) = 60,000 h$^{-1}$), accurately controlled by a gas distribution system with electric mass flow controllers (Brooks 5850 TR). The effluent was analyzed on-line by MKS-MultiGas

analyzer. The range of test temperature was set at 50 to 400 °C. Alkane conversion was calculated by the following equations:

$$X_{CnH2n+2}(\%) = \frac{[C_nH_{2n+2}]in - [C_nH_{2n+2}]out}{[C_nH_{2n+2}]in} \times 100\%, n \geq 1 \quad (2)$$

where $[C_nH_{2n+2}]_{in}$ and $[C_nH_{2n+2}]_{out}$ represented the inlet and outlet concentration of $C_nH_{2n+2}$, respectively; $S_{co2}$ stand for $CO_2$ selectivity; [CO], $[CO_2]$, and $[C_nH_n]$ represented the concentration of CO, $CO_2$, and $C_2H_4$ (or $C_3H_6$), respectively. The reaction temperatures for 10%, 50%, and 90% conversion of $C_nH_{2n+2}$ to $CO_2$ were assigned to $T_{10}$, $T_{50}$, and $T_{90}$, respectively. $Y_{CO2}$ stands for $CO_2$ yield.

The surface area normalized rate (μmol$_{C2H6}$ m$^{-2}$ s$^{-1}$) was calculated by the following equation:

$$r_{C2H6} = \frac{X_{C2H6} \cdot F_{C2H6}}{S_{BET} \cdot m_{cat.}} \quad (3)$$

where $X_{C2H6}$ (%) represents the conversion of $C_2H_6$, $F_{C2H6}$ (mol s$^{-1}$) is the mole flow rate of $C_2H_6$, $S_{BET}$ (m$^2$ g$^{-1}$) is the surface area of tested materials, $m_{cat.}$(g) is the mass of the applied catalyst.

## Computational details

All calculations performed in this work were within the framework of Density Functional Theory (DFT) by using the Vienna Ab initio simulation program (VASP) 6.1.0. The projector-augmented wave (PAW) pseudopotentials were used to describe the electron-ion interactions[52]. The generalized gradient approximation with the Perdew-Burke-Ernzerhof functional (GGA-PBE) was used to treat the electron exchange and correlation energy[53]. Electron smearing was employed via Gaussian smearing method with a smearing width consistent to 0.05 eV. Valence electrons were described by a plane wave basis with an energy cutoff energy of 450 eV. Optimized structures were obtained by minimizing the forces on each atom using the conjugate gradient (CG) algorithm until <0.03 eV/Å. The energy convergence criteria were set to 10$^{-5}$ eV. A correction for Coulomb and exchange interactions was employed by setting $U_{eff} = 3.5$ eV and 3.1 eV ($U_{eff}$ = coulomb U − exchange J) for Co and Mn atoms, respectively, using the model proposed by Dudarev et al.[54]. The D3 correction method (DFT-D3) was employed in order to include the van der Waals (vdW) interactions[55].

The formation energy of an oxygen vacancy ($E_{Ov}$) was calculated by the following equation:

$$E_{ov} = E_{slab,Ov} - E_{slab} + 1/2 E_{O2} \quad (4)$$

where $E_{slab,Ov}$ is the energy of the defective $MnCo_2O_4$ slab surface, $E_{slab}$ is the energy of the perfect slab surface, and $E_{O2}$ is the energy of the gaseous oxygen molecule.

The adsorption energy of oxygen ($E_{ads,O2}$) was calculated based on a perfect slab surface by the following equation:

$$E_{ads,O2} = E_{slab,O2} - E_{slab} - E_{O2} \quad (5)$$

where $E_{slab,O2}$ is the energy of $MnCo_2O_4$ slab surface covered by the oxygen molecule, $E_{slab}$ is the energy of the clean slab surface, and $E_{O2}$ is the energy of the gaseous oxygen molecule.

The adsorption energy of ethane ($E_{ads,C2H6}$) was calculated based on a perfect slab surface by the following equation:

$$E_{ads,C2H6} = E_{slab,C2H6} - E_{slab} - E_{C2H6} \quad (6)$$

where $E_{slab,C2H6}$ is the energy of $MnCo_2O_4$ slab surface covered by the ethane molecule, $E_{slab}$ is the energy of the clean slab surface, and $E_{C2H6}$ is the energy of the gaseous ethane molecule.

The calculated equation for the surface energy ($\Omega$) of the three crystal facets can be expressed as follows:

$$\Omega = 1/2A[G_{slab} - N(O)\mu(O) - N(Mn)\mu(Mn) - N(Co)\mu(Co)] \qquad (7)$$

where $G_{slab}$ is approximated as the total energy calculated by DFT and A is the surface area of the crystal facet terminations. $N(O)$, $N(Mn)$ and $N(Co)$ are the numbers of O, Mn and Co atoms on the crystal facets, and $\mu(O)$, $\mu(Mn)$ and $\mu(Co)$ are the chemical potentials of O, Mn and Co atoms. Since the chemical potentials of O, Mn and Co are assumed to be in equilibrium with the bulk $MnCo_2O_4$, they are related through the following expression:

$$\mu(MnCo_2O_4) = \mu(Mn) + 2\mu(Co) + 4\mu(O) \qquad (8)$$

where $\mu(MnCo_2O_4)$ is the chemical potential of the bulk $MnCo_2O_4$, which is approximated by the total energy of bulk $MnCo_2O_4$ unitary.

## Data availability

The data that generated in this study are provided in the published article and its Supplementary Information/Source Data files. The data that support the findings of this study have been deposited in the FigShare database under accession code https://doi.org/10.6084/m9.figshare.24150888. Source data are provided with this paper.

## Code availability

The data that generated in this study are provided in the published article and its Supplementary Information/Source Data file. The data that support the findings of this study have been deposited in the FigShare database under accession code https://doi.org/10.6084/m9.figshare.24150888. Source data are provided with this paper.

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

## Acknowledgements

This work is supported by the Liaoning Revitalization Talent Program (No. XLYC2008032), the Fundamental Research Funds for the Central Universities (Nos. DUT22LAB602 and DUT21LK22), the Technology Development Contract of Sinopec (Grant No. 322068), the National Natural Science Foundation of China (No. 22202028), the Fellowship of China Postdoctoral Science Foundation (No. 2022M720638) and the Talent Introduction Program of Postdoctoral International Exchange Program (Grant No. YJ20210238). Also, the authors would like to thank Dr. Jing Li (Application Specialist) from TESCAN CHINA for TOF-SIMS measurement. We would like to thank Dr. Jaming Wu (Dalian University of Technology) for creating the Featured Image for this paper.

## Author contributions

Haiyan Wang: Conceptualization, Investigation, Methodology, Formal analysis, Data curation, Writing—original draft, Software, Writing—review and editing, Visualization, Funding acquisition. Shuang Wang: DFT calculation, Methodology, Formal analysis, Data curation, Software, Writing—review and editing. Shida Liu: Conceptualization, Investigation, Methodology, Formal analysis, Data curation, Software, Writing—review and editing, Visualization, Funding acquisition, Supervision, Project administration. Yiling Dai: Data curation, Writing—review and editing. Zhenghao Jia: Data curation. Xuejing Li: Data curation. Feixiong Dang: Data curation. Shuhe Liu: Data curation. Kevin J. Smith: Writing—review and editing. Xiaowa Nie: Writing—review and editing, supervision. Shuandi Hou: Conceptualization, Supervision, Funding acquisition, Project administration. Xinwen Guo: Conceptualization, Supervision, Funding acquisition, Project administration.

## Competing interests

The authors declare no competing interests.
