## [Peer Review File · Nature Communications]

Redox-induced controllable engineering of MnO₂-Mn_xCo_{3-x}O₄ interface to boost catalytic oxidation of ethaneREVIEWER COMMENTS

Reviewer #1 (Remarks to the Author):

In this work, the authors report that a mixture of $\text{MnO}_2/\text{MnxCo}_3\text{-xO}_4$ shows high activity for C_2H_6 combustion. The authors studied a synergy between MnO_2 and $\text{MnxCo}_3\text{-xO}_4$ and conclude that C_2H_6 tends to be adsorbed on the interfacial Co sites and subsequently break the C-H bonds on the reactive lattice O on MnO_2 . Based on the following reasons, I do not find the manuscript suitable for publication on a high-level Journal as Nature Communications.

1. The $\text{MnO}_2/\text{MnxCo}_3\text{-xO}_4$ mixture was prepared by a conventional preparation method of MnCoOx catalysts when the Mn/Co ratio was 0.5. MnCoOx catalysts have been reported for various catalytic reactions, especially for combustion of hydrocarbons and VOCs. The C_2H_6 combustion by a MnCoOx catalyst in the present work for is a logical extension of the previous works.
2. From environmental viewpoint, C_2H_6 combustion is less important than CH_4 combustion. Due to low importance of C_2H_6 combustions, number of the catalyst reported for C_2H_6 is too much less than that for CH_4 . If the MnCo catalyst showed much higher activity for CH_4 combustion than Pd/ Al_2O_3 catalysts, the originality of this work should be high.
3. The important scientific part, that is the synergy between MnO_2 and $\text{MnxCo}_3\text{-xO}_4$, lacks scientific evidences. Kinetic and structural experiments with $\text{MnxCo}_3\text{-xO}_4$ are not shown as a comparative purpose. The $\text{MnO}_2/\text{MnxCo}_3\text{-xO}_4$ shows high activity for C_2H_6 combustion than $\text{MnO}_2/\text{MnCo}_2\text{O}_4$. However, only MnCo_2O_4 is utilized as a comparative sample for kinetic and structural experiments.

Reviewer #2 (Remarks to the Author):

The interface engineering among various components in catalysts plays an essential role in designing high-performance catalysts. In general, the metal-support interface has been widely investigated, while the insight understanding on oxide-oxide interface is relatively rare. In the present work, the authors reported the manipulation of the $\text{MnO}_2\text{-MnxCo}_3\text{-xO}_4$ interface through changing the Mn/Co ratio of MnCoOx catalysts or adjusting the annealing conditions, and they elucidated the synergistic effects between the two that can enhance both the ethane adsorption/activation and the lattice oxygen mobility. These findings, in my opinion, will provide valuable insights for the rational design of efficient catalysts for alkane combustion. This reviewer recommends the acceptance of the work after addressing the following issues.

1. The manuscript suggests that MnCoOx catalysts create the $\text{MnO}_2\text{-MnxCo}_3\text{-xO}_4$ interface by forming the MnO_2 layer. The solid evidence on the formation of the $\text{MnO}_2/\text{MnCo}_2\text{O}_4$ interface should be enhanced. Are there new species formed due to the interaction between the two oxides?
2. The O1s XPS peak in Fig. 1e was not well fitted. In addition, or the O1s XPS analysis, the authors stated that the lower O_{latt} B.E., the weaker interaction between M and O. Do they have any supports? It is controversial to assign the fitted peaks to Co^{2+} and Co^{3+} , so related references and reasonable analysis of the satellite peaks is necessary.
3. It was observed that the $\text{MnO}_2\text{-MnxCo}_3\text{-xO}_4$ interface is induced by redox treatment, while the catalyst also shows high stability during the long-term reaction. The evolutions of the interface during the redox treatment and the long-term reaction are worthy to comparing.
4. I know that a reasonable modelling for such a complex catalyst system is very difficult in density functional theory calculation. The reviewer note that the $\text{MnxCo}_3\text{-xO}_4$ is nonstoichiometric. In this case, how to model $\text{MnxCo}_3\text{-xO}_4$?
5. It seems that the interfacial Co sites should be the active sites, but I am interested in the effect of Co species on the MnO_2 . The roles of Co and Mn species are expected for more discussions.

Reviewer #3 (Remarks to the Author):

The authors studied catalytic ethane oxidation over $\text{MnO}_2\text{-Co}_2\text{O}_3$ -based mixed oxide materials. To do this work, the authors synthesize the material by chemical reduction method, which facilitates

lattice oxygen on the surface and further enhances the reactivity. I would gladly recommend its publication; however, it needs to be further improved to be published in Nature Communications.

1. The authors need to provide the motivation and the importance of the catalytic hydrocarbon oxidation along with the interfacial engineering materials.
2. According to the experiment for the MnCoOx synthesis, the authors mixed KMnO₄ with cobalt nitrate precursor. However, it is not clear how the authors employed a redox-controlled synthesis method. Moreover, the experimental details for the synthesis are further required. In this regard, the authors need to provide i) which solvent and concentration the author used for the precursors, ii) how the author controlled the synthesis, for example, pH during the synthesis and aging time, and iii) how the authors are sure K is completely removed during the synthesis and post-treatment.
3. The Raman spectra for the synthetic materials is helpful to understand the structural changes depending on the composition. However, I kindly ask the authors to provide the peak assignment for all the peaks in the spectra, not only stop to the main peak at 670 cm⁻¹ and also more discussion about those peaks according to the compositions.
4. For TOF calculation, the author normalized the reaction rate by metal sites quantified by CO chemisorption. However, if lattice oxygen can be a descriptor for the oxidation reaction and the reaction rate does not rely on the metal sites, the author needs to normalize the rate by the surface or lattice (active) oxygen and then figure out the active site compared to the metal sites.
5. In addition to comment #4, it questions whether the author can provide a quantification of active oxygen for ethane oxidation from O₂-TPD, which only shows the area of the specific peak.
6. By considering the apparent activation energy associated with the sensitivity of the materials to the reaction temperature, there seems to be a conflict in the reactivity. For example, MnCoOx-0.5 has higher sensitivity to the temperature observed in the light-off ethane profile. Moreover, it shows two different slopes below 20% ethane conversion in the light-off curves. Hence, the authors should describe those different slope changes and then link them to the reactivity.
7. For oxidation reactivity depending on the chain length of hydrocarbon, it is unclear how the reactivity is limited by the initial H abstraction.
8. To deconvolute the profile for ethane-TPRS, the authors also need to show MnCo₂O₄ to ensure the peak at 207 °C attributed to MnO₂-MnCo₂O₄.
9. The authors argued that the higher selectivity C₁₆O₂ and C_{16/18}O₂ higher than C₁₈O₂ from TAP indicates the direct participation of lattice O located at MnO₂-MnCo₂O₄ interface. However, it is very hard to find out how the authors reached out this conclusion from the TAP experiment and the oxygen scrambling result. It has questions to prove the interfacial sites for the lattice oxygen in this manuscript.

RESPONSE TO REVIEWERS' COMMENTS

We thank the reviewers for the constructive comments and suggestions to help us improve the quality of the manuscript. We have addressed all the comments point-by-point and revised the manuscript accordingly. We hope that the detailed responses and changes we have made to the manuscript titled “*Redox-induced controllable engineering of MnO₂-Mn_xCo_{3-x}O₄ interface to boost catalytic oxidation of ethane*” now make it acceptable for *Nature Communications*.

In this response letter, comments from the reviewers are summarized in black typeface with the original comments quoted in *italic*, and our responses are in blue typeface. All major changes have been highlighted in yellow in the main text. We have attached a clean version of the revised manuscript and a file with tracking changes.

A point-to-point response to the comments from three reviewers

Reviewer #1

In this work, the authors report that a mixture of MnO₂/Mn_xCo_{3-x}O₄ shows high activity for C₂H₆ combustion. The authors studied a synergy between MnO₂ and Mn_xCo_{3-x}O₄ and conclude that C₂H₆ tends to be adsorbed on the interfacial Co sites and subsequently break the C-H bonds on the reactive lattice O on MnO₂. Based on the following reasons, I do not find the manuscript suitable for publication on a high-level Journal as Nature Communications.

Comment 1: *The MnO₂/Mn_xCo_{3-x}O₄ mixture was prepared by a conventional preparation method of MnCoO_x catalysts when the Mn/Co ratio was 0.5. MnCoO_x catalysts have been reported for various catalytic reactions, especially for combustion of hydrocarbons and VOCs. The C₂H₆ combustion by a MnCoO_x catalyst in the present*

work for is a logical extension of the previous works.

Reply: In this study, we have prepared the MnCoO_x catalysts by carefully engineering the chemical reduction process of Mn and Co salts via varying Mn/Co ratios and annealing conditions, eventually a catalytic system with abundant $\text{MnO}_2\text{-Mn}_x\text{Co}_{3-x}\text{O}_4$ interfaces was created. A detailed description of the preparation methodology was provided in the revised Supporting Information. Our results showed that the obtained $\text{MnCoO}_{x-0.5}$ catalyst exhibited an exceptional performance in ethane oxidation compared to other reported catalysts (Table S6), which incur us to explore the scientific questions behind and figure out the “structure-activity” relationship. Even though there are studies of MnCoO_x catalysts for VOC treatment, most of them often take surface lattice O as active sites and lack deep mechanistic investigation. Instead, our study specifically focused on exploring the catalytic effects of oxide-oxide interface, a topic that has been rarely discussed. We identified the synergistic effect between MnO_2 and $\text{Mn}_x\text{Co}_{3-x}\text{O}_4$ as well as their interfacial effects, and proposed the reaction mechanism accordingly based on a combination of experimental and computational approaches. Additionally, we established a correlation between interfacial sites ($\text{MnO}_2\text{-MnCo}_2\text{O}_4$) and oxidation activity to shed light on their intrinsic properties. Thus, we believe this work could deepen the fundamental understanding on the catalytic effects of interfacial sites for VOC combustion other than the present publication.

Comment 2: *From environmental viewpoint, C_2H_6 combustion is less important than CH_4 combustion. Due to low importance of C_2H_6 combustions, number of the catalyst reported for C_2H_6 is too much less than that for CH_4 . If the MnCo catalyst showed much higher activity for CH_4 combustion than $\text{Pd/Al}_2\text{O}_3$ catalysts, the originality of this work should be high.*

Comment 2-1: *From environmental viewpoint, C_2H_6 combustion is less important than CH_4 combustion. Due to low importance of C_2H_6 combustions, number of the catalyst reported for C_2H_6 is too much less than that for CH_4 .*

Reply: We thank the reviewer for pointing this out. Additional statements have been

made in the *Introduction* of revised manuscript to stress the importance of studying ethane combustion as well as the feature of synthesized MnCoO_x interfacial catalysts. In fact, there are some reasons for us to choose ethane oxidation as a probe reaction. Firstly, it is well-known that ethane is a major byproduct from coal gasification, rock oil outgassing, and various chemical production processes (such as epoxidation of ethylene etc.). As a major source of volatile organic compound (VOC), the emission of ethane could cause serious environmental issues. Also, the inherently strong C-H bonds ($\Delta E_{\text{C-H}}=442 \text{ kJ mol}^{-1}$) that involved in ethane molecules have made it thermodynamically stable, leading to the activation of ethane a crucial step. Thereby, the scientific investigation of ethane oxidation over certain materials seems to be necessary. For example, the combustion of ethane may produce some unoxidized or partially oxidized species (methane and ethylene) due to the incomplete oxidation, which depends on the properties of the synthesized catalyst.

Compared to ethane, methane is a common emission resulting from many activities, such as agriculture, transportation, and manufacturing. However, methane is excluded from VOC monitoring in many cases, as it is relatively non-toxic and poses little harm to human health. By excluding methane from VOC analysis and detection, it is easier to understand the impact of non-methane volatile organic compounds (NMVOCs). Ethane as one of the representative compounds of NMVOC, should be given more attention based on the stringent standards on VOC emissions. Also, ethane is an inert alkane that vastly exists in nature gas (accounts about 10%). During the catalytic combustion of nature gas, the effects of ethane oxidation should be considered in order to assess the energetic performance of combustion process. Aside from this, the presence of ethane may affect the oxidation of states of the applied catalyst and disturb the surface oxygen layer, thus influencing the activity and/or selectivity to CO_2 . Hence, a fundamental study on the combustion of ethane over certain materials could offer more insights into the specific mechanism and the underlying structure-activity relationship. Up to now, few studies were reported about ethane catalytic combustion. Therefore, more efforts should be devoted in this research area, especially on the

mechanistic exploration of ethane combustion, to provide guidance for the development of catalysts with superior performance for light alkanes.

Comment 2-2: If the MnCo catalyst showed much higher activity for CH₄ combustion than Pd/Al₂O₃ catalysts, the originality of this work should be high.

Reply: To address the reviewer's question, we examined the catalytic performance of CH₄ combustion over MnCoO_x and 1Pd/Al₂O₃ catalysts at the same reaction conditions (ca. 200 mg catalyst, [CH₄]=2000 ppm, Q=200 mL min⁻¹, WHSV=30,000 h⁻¹). Noted that, the MnCoO_x exhibited higher activity than 1Pd/Al₂O₃ reference (**Figure R1, Table R1**). Considering the effectiveness of Pd on the cleavage of C-H bonds, we further employed MnCoO_x as substrate to disperse Pd (denoted as xPd/MnCoO_x). After that, we compared the performance of xPd/MnCoO_x catalyst with Pd/Al₂O₃ reference in methane combustion. Apparently, methane was easily activated on the 0.2Pd/MnCoO_x catalyst due to the presence of Pd. Compared to MnCoO_x, the ignite temperature (T₁₀) significantly decreased to 230 °C on 0.2Pd/MnCoO_x catalyst, which demonstrates the effectiveness of Pd on C-H bond cleavage. Also, it was found that the T₉₀ of 1.0Pd/MnCoO_x (311 °C) was remarkably lower than that of 1Pd/Al₂O₃ (435 °C), perhaps due to the good distribution of Pd over MnCoO_x support. As part of an independent study, a thorough examination is presently undertaken to investigate the effects of MnCoO_x on Pd. We expect that our ongoing research would provide more insights into the interaction between Pd and MnCoO_x by using methane as a probe molecule. However, the present work in this manuscript aimed at recognizing the interfacial effects of different oxides, specifically the synergistic interaction between MnO₂ and Mn_xCo_{3-x}O₄, by taking ethane oxidation as a probe reaction.

Figure R1. CH₄ conversion of the as-synthesized catalysts.

Table R1. Catalytic performance of prepared x Pd/MnCoO _{x} catalysts and 1Pd/Al₂O₃ reference.

Catalyst	CH ₄ oxidation		
	T ₁₀ (°C)	T ₅₀ (°C)	T ₉₀ (°C)
MnCoO _{x}	246	286	342
0.2Pd/MnCoO _{x}	230	272	318
1.0Pd/MnCoO _{x}	213	260	311
1.0Pd/Al ₂ O ₃	274	370	435

Reaction conditions: ca. 200 mg catalyst, [CH₄] = 3000 ppm, Q=200 mL min⁻¹, WHSV=60,000 h⁻¹.

Comment 3: The important scientific part, that is the synergy between MnO₂ and Mn _{x} Co_{3- x} O₄, lacks scientific evidences. Kinetic and structural experiments with Mn _{x} Co_{3- x} O₄ are not shown as a comparative purpose. The MnO₂/Mn _{x} Co_{3- x} O₄ shows

high activity for C₂H₆ combustion than MnO₂/MnCo₂O₄. However, only MnCo₂O₄ is utilized as a comparative sample for kinetic and structural experiments.

Reply: We appreciated the critical comments from the reviewer. To address the concern of the reviewer, we have further designed new sets of experiments as well as performed additional DFT calculations to illustrate the synergistic effects between MnO₂ and Mn_xCo_{3-x}O₄ in ethane oxidation. The new evidences have been added in the revised manuscript.

For the experimental part, aside from the MnCo₂O₄ catalyst, we prepared other catalysts as comparative samples for kinetic and structural experiments, including Phy_MnCo₂O₄-MnO₂, LP_MnCo₂O₄-MnO₂, and MnO₂. As shown in **Figure R2** and **Table R2**, the data show that MnO₂ alone gives the lowest activity and only 50% of ethane could be converted at 350 °C (T₅₀=350 °C) under conditions of 3000 ppm ethane, Q=200 mL min⁻¹, and WHSV of 60,000 h⁻¹. Interestingly, the low-temperature activity is significantly enhanced once MnCo₂O₄ has been added into MnO₂ regardless of their mixing methods. Note that, the temperature of T₅₀ dramatically reduced to 304 °C over the physically mixed MnO₂ and MnCo₂O₄ catalyst (denoted as Phy_MnCo₂O₄-MnO₂), indicating the synergistic effects between MnO₂ and MnCo₂O₄. Moreover, the up-down layer packed MnCo₂O₄-MnO₂ catalyst (denoted as LP_MnCo₂O₄-MnO₂) also exhibits a comparable performance (T₅₀=311 °C) for ethane conversion. However, the catalytic activity of Phy_MnCo₂O₄-MnO₂ (T₅₀=304 °C) and LP_MnCo₂O₄-MnO₂ (T₅₀=311 °C) catalysts are all lower than that of MnO_x/MnCo₂O₄ model catalyst (T₅₀=270 °C), indicating the significant role of the interfacial sites due to the intimacy between these two oxides. Aside from the formation of metal oxide/metal oxide interfaces, the underlying MnCo₂O₄ substrate could affect the electron distribution of MnO₂ (Figure 6b, Figure S32), therefore the catalytic performance of MnO₂/MnCo₂O₄ catalyst is remarkably distinct from that of the Phy_MnCo₂O₄-MnO₂ and LP_MnCo₂O₄-MnO₂ catalysts. The results of newly added experiments have been included in the revised Supporting Information, specifically in Figure S25.

Figure R2. Catalytic performance of prepared MnCoO_x catalysts and references in ethane combustion.

Table R2. Catalytic performance of prepared MnCoO_x catalysts and references.

Catalyst	C ₂ H ₆ oxidation		
	T ₁₀ (°C)	T ₅₀ (°C)	T ₉₀ (°C)
MnO ₂	249	350	N/A
LP_MnCo ₂ O ₄ -MnO ₂	241	311	392
Phy_MnO ₂ -MnCo ₂ O ₄	235	304	383
MnCo ₂ O ₄	211	291	349
MnO _x /MnCo ₂ O ₄	205	270	317
MnCoO _{x-0.5}	175	205	220

A macroscopic kinetics studies (**Figure R3**) exhibited that the apparent activation energy (E_a) of ethane conversion to CO₂ over the Phy_MnO₂-MnCo₂O₄ ($E_a=93.5\pm 1.4$ kJ mol⁻¹) and LP_MnCo₂O₄-MnO₂ ($E_a=97.7\pm 0.7$ kJ mol⁻¹) catalyst are obviously lower than that over MnO₂ ($E_a=109.9\pm 2.9$ kJ mol⁻¹), suggesting that the synergy between MnO₂ and MnCo₂O₄ shows the potential of creating a new reaction path to facilitate the oxidation of ethane. While the E_a of both mixtures are higher than that over MnO_x/MnCo₂O₄ ($E_a=87.9\pm 2.2$ kJ mol⁻¹), further demonstrating the essential role of MnO_x-MnCo₂O₄ interfaces. Therefore, we can conclude that the interfacial sites that

established by the close contact between MnO₂ and MnCo₂O₄ exert a significant influence on ethane oxidation. The relevant explanations have been added into the revised manuscript on page 13.

Figure R3. The Arrhenius plots of various MnCo-based catalysts and references.

Apart from the kinetic experiments, DRIFT analysis was conducted to track the reaction intermediates. The DRIFT spectra were recorded after 10 min injection of the feed gas in the temperature range of 50-200 °C (**Figure R4**). Apparently, the peak intensity of C-H stretching vibrations (ca. 3000 cm⁻¹) of ethane decreased with increased temperature, indicating the activation of ethane at ≥ 100 °C over MnCoO_{x-0.5}. Various oxygenates were generated as indicated by the formation of multiple adsorption bands at 1000-1700 cm⁻¹. Note that, a sharp peak appeared at 1458 cm⁻¹, which can be assigned to acetate (CH₃COO*) from H abstraction of ethane on lattice O*. The peak intensity of CH₃COO* decreased with increased temperature from 50 to 150 °C, indicating their further conversion to other oxygenates. While the amount of CH₃COO* slightly increased above 150 °C, perhaps due to the participation of MnO₂-Mn_xCo_{3-x}O₄

interface, thus producing more active sites for C₂H₆ activation. The C₂H₆-TPSR results (Figure 4a) also demonstrate the formation of MnO₂-Mn_xCo_{3-x}O₄ interface. Moreover, the peaks at 1065, 1120, and 1172 cm⁻¹ can be assigned to CH₃O*, which is likely due to the breakage of C-C bond of CH₃COO*. The intensity of CH₃O* gradually decreased with increased temperature because of their further conversion. Aside from these, other dehydrogenation products were detected as well, such as carboxylate acid (HCO*: 1327 & 1514 cm⁻¹), formate (HCOO*: 1603 cm⁻¹), bicarbonate (HCO₃*: 1403 cm⁻¹), and carbonate (CO₃²⁻: 1303 & 1365 cm⁻¹). Overall, this result suggests that there are more O-intermediates involved in ethane oxidation via Mars van Krevelen (MvK) mechanism. Ultimately, we can gain a preliminary understanding of the reaction mechanism of ethane combustion over the MnCoO_x-0.5 catalyst (C₂H₆ → C₂H₆-O* → CH₃CH₂O* → CH₃HCOO* → CH₃O* & HCOO* → HCOO*, HCO*, HCO₃*, CO₃²⁻ → CO₂, H₂O). The revised manuscript presents the *in-situ* DRIFT results on page 16, and further interpretation can be found in the Supporting Information of Figure S39.

Figure R4. *In-situ* DRIFT spectra of C₂H₆ oxidation over the MnCoO_x-0.5 catalyst.

Furthermore, to gain a deeper understanding on the MnCoO_x interfacial system,

additional DFT calculations were carried out over the established MnCO₂O₄/MnO₂-111-C model based on the above-proposed reaction path. The calculated energy profiles and the corresponding structures of all states involved are shown in **Figures R5 and R6**. After dissociating the first C-H bond of C₂H₆, the formed CH₃CH₂^{*} species is prone to bond on Co sites that located at the interface of MnO₂ and MnCO₂O₄, which aligns with the C₂H₆-TPSR results. Started with CH₃CH₂^{*}, there are two possible routes for the breakage of second C-H bond: i) the adsorbed CH₃CH₂^{*} is further dissociated to form CH₃CH^{*} on the same site (as shown in **Figure R5: c'**); ii) the adsorbed CH₃CH₂^{*} changes its adsorption site and migrates to the lattice O^{*} of upper MnO₂ cluster to form CH₃CH₂O^{*} (as in **Figure R5: c**). The calculation results show that the formation of CH₃CH₂O^{*} is thermodynamically much more favorable by releasing an energy of 2.14 eV. In contrast, the formation of CH₃CH^{*} is endothermic by 1.40 eV, which is thermodynamically unfavorable. Therefore, the CH₃CH₂O^{*} is regarded as a preferred intermediate. These calculation results are in line with the *in-situ* XPS results, which imply that the further dehydrogenation mostly occurs on the upper MnO₂ cluster. After that, the generated CH₃CH₂O^{*} species continues to dehydrogenate, either with the neighboring Co sites to form Co-H (**Figure R5: d'**) or with ^{*}OH to form H₂O and O vacancy (^{*}), ultimately leading to the formation of the CH₃HCOO^{*} intermediate (**Figure R5: d**). Based on the energy diagram in **Figure R5**, it is noticeable that the formation of CH₃HCOO^{*} is energetically more favorable than the formation of CH₃CHO^{*}. In the next step, the CH₃HCOO^{*} is decomposed into CH₃O^{*} and HCOO^{*} by breaking the C-C bond, releasing an energy of 1.8 eV (**Figure R5: e**). After the C-C dissociation of CH₃HCOO^{*}, two paths were considered for HCOO^{*} conversion (**Figure R6: f, f'**). By comparison, the continuous dehydrogenation of HCOO^{*} on the upper MnO₂ cluster is thermodynamically more favorable, resulting in the formation of CO₂. Subsequently, a continuous dehydrogenation of CH₃O^{*} species occurs, sequentially forming CH₂O^{*}, CHO^{*}, CO₂^{*}, and CO₂(g). The optimal energy path for ethane oxidation on the MnCO₂O₄/MnO₂-111-C catalyst surface is illustrated in **Figure 6**. Comparing the two C-H bond dissociation steps, our calculation results reveal that the

1st H abstraction is the rate-determining step on the MnCo₂O₄/MnO₂ interfacial model, with a barrier of 0.89 eV. Overall, the combination of *in-situ* DRIFTS experiments with DFT calculations identify the underlying reaction mechanism of ethane oxidation and provide more scientific insights into the synergistic effects between MnO₂ and Mn_xCo_{3-x}O₄. The revised manuscript presents the DFT results on pages 15-16, and Figure 6 was updated by including the Fig.6f into the compiled figures.

Figure R5. Energy diagrams of partial elementary reactions involved in ethane oxidation on the MnCo₂O₄/MnO₂-111-C catalyst surface and the optimized structures of all species involved.

Figure R6. Energy diagram of the optimal reaction path for ethane oxidation on the MnCo₂O₄/MnO₂-111-C catalyst surface and the optimized structures of all species involved.

Reviewer #2:

The interface engineering among various components in catalysts plays an essential role in designing high-performance catalysts. In general, the metal-support interface has been widely investigated, while the insight understanding on oxide-oxide interface is relatively rare. In the present work, the authors reported the manipulation of the MnO₂-Mn_xCo_{3-x}O₄ interface through changing the Mn/Co ratio of MnCoO_x catalysts or adjusting the annealing conditions, and they elucidated the synergistic effects between the two that can enhance both the ethane adsorption/activation and the lattice oxygen mobility. These findings, in my opinion, will provide valuable insights for the rational design of efficient catalysts for alkane combustion. This reviewer recommends the acceptance of the work after addressing the following issues.

RESPONSE:

We thank the reviewer for the overall positive reviews on our papers. We also thank the

reviewer for the constructive comments. We have revised the manuscript by incorporating all of Reviewer 2's comments, please see the replies below as well as the yellow highlighted areas in the revised manuscript.

Comment 1: The manuscript suggests that $MnCoO_x$ catalysts create the MnO_2 - $Mn_xCo_{3-x}O_4$ interface by forming the MnO_2 layer. The solid evidence on the formation of the $MnO_2/MnCo_2O_4$ interface should be enhanced. Are there new species formed due to the interaction between the two oxides?

Reply: We thank the reviewer for this suggestion. In order to get more insights of the $MnO_2/MnCo_2O_4$ interface, we performed a depth profile analysis by XPS on the $MnCoO_x$ catalyst. A depth profile of Co^{2+}/Co^{3+} and Mn^{4+}/Mn^{3+} atomic ratios was obtained from analyzing the XPS data (Figure R7). Apparently, the ratio of Mn^{4+}/Mn^{3+} dramatically decreased with increasing etching time in the first 60 s (equivalent to $60\text{ s} \times 0.08\text{ nm/s} = 4.8\text{ nm}$), and reached to a relatively stable state afterwards. Accordingly, the ratio of Co^{2+}/Co^{3+} slowly increased with increased etching time in the first 60 s. Overall, the XPS analysis suggest that there are more Mn^{4+} species (MnO_2) retain on the surface of $MnCoO_x$ catalyst. The thickness of MnO_2 is about 4.8 nm, which is consistent with what we observed from HRTEM images (Figure 3b).

Figure R7. XPS depth profile analysis of the $MnCoO_{x-0.5}$ catalyst.

After that, TEM images were performed to compare the structures of the pristine and etched $\text{MnCoO}_x-0.5$ catalyst (**Figure R8**). Clearly, we identified the existence of MnCo_2O_4 spinel and MnO_2 from measuring the distance of observed lattice fringes. It was found that the d-spacing of pristine MnCoO_x catalyst is about 0.48 nm, which corresponds to the (111) plane of MnCo_2O_4 . Few MnO_2 layers with a d-spacing of 0.21 nm was found on the surface of pristine MnCo_2O_4 , as shown on the upper panel of **Figure R8(a)**. While, the etched MnCoO_x showed a similar morphology as that of the pristine sample, but with less MnO_2 layers on MnCo_2O_4 nanosheet (**Figure R8(b)**). Apart from MnCo_2O_4 and MnO_2 , no new phases were detected. Therefore, both the XPS depth profile and the associated TEM images can be used as new evidences to support the formation of MnO_2 - MnCo_2O_4 interface. These mentioned results have been included in the revised Supporting Information, specifically in **Figure S17**.

Figure R8. TEM images of $\text{MnCoO}_x-0.5$ catalyst: (a) pristine; (b) Ar^+ sputter etched.

Comment 2: The O1s XPS peak in Fig. 1e was not well fitted. In addition, or the O1s XPS analysis, the authors stated that the lower O_{lat} B.E., the weaker interaction between M and O. Do they have any supports? It is controversial to assign the fitted peaks to Co^{2+} and Co^{3+} , so related references and reasonable analysis of the satellite peaks is necessary.

Comment 2-1: The O1s XPS peak in Fig. 1e was not well fitted.

Reply: In response to the reviewer’s suggestion, we undertook the process of re-deconvoluting the O1s XPS peak to enhance the quality of its fitting, as presented in **Figure R9**. The corresponding parameters adopted for O1s peak splitting were reported in **Table R3**. The new fitting results have been updated in the revised manuscript on page 22, as shown in Figure 1e.

Figure R9. O1s XPS spectra of MnCoO_x catalysts.

Table R3. O1s peak fitting of MnCoO_x catalysts.

Catalysts	O _α (B.E.=530.1 eV)	O _β (B.E.=531.3 eV)	O _γ (B.E.=532.7 eV)
MnCoO _x -0.1	75.3%	13.5%	11.2%
MnCoO _x -0.5	86.0%	14.0%	0.0%
MnCoO _x -2.0	82.7%	13.4%	3.9%
MnO ₂	77.9%	16.5%	5.6%

Comment 2-2: In addition, or the O1s XPS analysis, the authors stated that the lower

Olatt B.E., the weaker interaction between M and O. Do they have any supports?

Reply: In the revised manuscript, we have cited a reference paper (*ACS Catal.* **2018**, *8*, 3435-3446, see Page 12 - ref [42])¹ to support our statement “the lower O_{latt} B.E., the weaker interaction between M and O”. It is generally accepted that a weaker interaction between M and O could lead to an increase in oxygen vacancies, thus increasing the electron density of lattice oxygen and resulting in a decrease in the B.E. of lattice oxygen.

Comment 2-3: *It is controversial to assign the fitted peaks to Co^{2+} and Co^{3+} , so related references and reasonable analysis of the satellite peaks is necessary.*

Reply: As suggested by the reviewer, a reasonable analysis on the satellite peaks of Co^{2+} and Co^{3+} species was performed based on several references (*ACS Catal.* **2019**, *9*, 7548-7567; *J. Am. Chem. Soc.* **2013**, *135*, *22*, 8283-8293; *ACS Appl. Mater. Interfaces* **2014**, *6*, 2439-2449; *J. Catal.* **2017**, *352*, 282-292; *J. Catal.* **2021**, *404*, 400-410).²⁻⁶ The peak deconvolution results of Co2p were presented in **Figures R10 & 11**. From **Figure R10**, we can see that the Co2p_{3/2} can be divided into four sets of components: one peak at B.E. = 780.2 eV is assigned to the surface Co^{3+} , and the other one at B.E.=781.7 eV is ascribed to the surface Co^{2+} species, and the other two peaks are assigned to the shakeup satellites. It is well-known that the cobalt oxide typically consists of Co^{3+} and Co^{2+} ions. The presence of satellite peaks suggests the partial reduction from Co^{3+} to Co^{2+} on the prepared MnCoO_x catalysts, indicating the coexistence of Co^{2+} and Co^{3+} on the prepared samples. In the revised manuscript (refer to Page 6), we have revised the method of peak deconvolution for Co2p spectra and conducted an analysis on the satellite peaks of Co2p.

Figure R10. Detailed peak deconvolution of Co 2p spectra of MnCoO_{x-0.5}.

Figure R11. XPS analysis of Co 2p spectra of MnCoO_x catalysts.

Comment 3: *It was observed that the MnO_2 - $Mn_xCo_{3-x}O_4$ interface is induced by redox treatment, while the catalyst also shows high stability during the long-term reaction. The evolutions of the interface during the redox treatment and the long-term reaction are worthy to comparing.*

Reply: We thank the reviewer for pointing this out. As described in the catalyst preparation procedure (refer to Supporting Information on page 3), the precipitate of $MnCoO_x$ was obtained by a redox-controlled chemical reduction process. Details have been listed in Table S10 of the revised Supporting Information. Subsequently, the annealing of the $MnCoO_x$ precipitates was conducted in air to facilitate the formation of MnO_2 - $MnCo_2O_4$ interfaces by taking advantages of the strong affinity of Mn species towards O_2 , eventually reached to a relatively stable state. After that, the catalytic performance of the resulting $MnCoO_x$ catalysts was assessed for ethane oxidation under O_2 -rich conditions. The cyclic stability tests showed that there is no deactivation on catalytic activity, indicating the good stability of $MnCoO_x$ catalysts in ethane oxidation. Furthermore, a long-term stability test was performed, which also proves the outstanding stability of $MnCoO_x$ catalysts. Therefore, both sets of experiments suggest that there is no structural evolution of the obtained $MnCoO_x$ catalysts after redox treatment.

Comment 4: *I know that a reasonable modelling for such a complex catalyst system is very difficult in density functional theory calculation. The reviewer note that the $Mn_xCo_{3-x}O_4$ is nonstoichiometric. In this case, how to model $Mn_xCo_{3-x}O_4$?*

Reply: We thank the reviewer for this insightful question. We referred to the $Mn_xCo_{3-x}O_4$ notation to represent the spinel phase in the obtained $MnCoO_x$ samples as it can be challenging to accurately determine the Mn and Co content in spinels due to the structural complexity of synthesized catalysts. Based on our experimental results, we chose the stoichiometric $MnCo_2O_4$ as the underlying spinel phase for our DFT calculation. More specifically, the HRTEM images clearly indicate the formation of

MnCo₂O₄ spinel mainly exposed with (111) facets on the synthesized MnCoO_x catalyst. Also, the lattice parameter of the optimized inverse MnCo₂O₄ model is about 8.14 Å, which is consistent with the XRD results. Therefore, we took MnCo₂O₄-111 as the underlying substrate and subsequently built the MnO₂-MnCo₂O₄ interfacial model to assess its catalytic behavior in ethane oxidation.

Comment 5: It seems that the interfacial Co sites should be the active sites, but I am interested in the effect of Co species on the MnO₂. The roles of Co and Mn species are expected for more discussions.

Reply: We thank the reviewer for this valuable comment, and additional discussions have been included in the revised manuscript on pages 14-15 along with Supporting Information in Figures S34-35, highlighting the distinct roles of Mn and Co species in the reaction. To better answer the reviewers' question, a MnO₂ supported Co-based catalyst (denoted as Co/MnO₂) was prepared and tested in ethane oxidation. As shown in **Figure R12.**, the ethane conversion of Co/MnO₂ catalyst is higher than that of MnO₂ at 300 °C (Co/MnO₂: conv.=78%; MnO₂: conv.%=26%), indicating the positive role of Co on ethane activation. Then, the C₂H₆-TPD was carried out to study the chemisorption behavior of C₂H₆ over the Co/MnO₂ catalyst. It was found that most of the adsorbed C₂H₆ was readily converted into CO at the onset of the TPD process of 50 °C over Co/MnO₂ catalyst, which may result from the strong C-C bond breakage ability over Co-related sites. While, the rest of adsorbed C₂H₆ was converted into CO₂ and H₂O above 300 °C. Differently, for the pure MnO₂, most of the C₂H₆ was mainly desorbed as C₂H₆ and CO₂ with increased temperature. Also, through calculating the integrated area of C-related species, we found that the amount of adsorbed C₂H₆ was higher on the Co/MnO₂ catalyst compared to the pure MnO₂, perhaps due to the formation of Co-O-Mn bonds at the interface between Co₃O₄ and MnO₂, thus facilitating the adsorption of C₂H₆ as well as increasing the O mobility at interfacial areas.

Figure R12. (a) Light-off curves; (b) catalytic performance of Co/MnO₂ and MnO₂ in ethane oxidation; (c) C₂H₆-TPD over Co/MnO₂ catalyst; (d) C₂H₆-TPD over MnO₂.

To get more insights of the Co-O-Mn sites, additional DFT calculations were performed to investigate the C₂H₆ adsorption on the Co/MnO₂ catalyst (**Figure R13**). Bulk MnO₂ models exposed with (111), (110), and (101) planes were built to correlate with what we observed from HRTEM images. Furthermore, the exposed Mn sites were replaced by Co atoms (n=1-2) to explore their influence on the ethane adsorption. Note that, the adsorption strength of ethane increased with increasing the Co substitution amount regardless of the exposed plane of MnO₂, suggesting a promoting effects of Co sites on the C₂H₆ adsorption.

Overall, both the experimental and DFT results could support the positive role of Co sites on MnO₂ in ethane oxidation.

Figure R13. The calculated adsorption energies of ethane on Co-doped MnO₂ surfaces and their optimized configurations.

Reviewer #3:

The authors studied catalytic ethane oxidation over MnO₂-Co₂O₃-based mixed oxide materials. To do this work, the authors synthesize the material by chemical reduction method, which facilitates lattice oxygen on the surface and further enhances the reactivity. I would gladly recommend its publication; however, it needs to be further improved to be published in Nature Communications.

RESPONSE:

We thank the reviewer for the overall positive reviews on our papers. We also thank the Reviewer for the constructive comments. We have revised the manuscript by incorporating all of Reviewer 3's comments, please see the replies below as well as the yellow highlighted areas in the revised manuscript.

Comment 1: *The authors need to provide the motivation and the importance of the catalytic hydrocarbon oxidation along with the interfacial engineering materials.*

Reply: We thank the reviewer for this valuable suggestion. As suggested by the

reviewer, we highlighted the importance of studying the catalytic hydrocarbon oxidation and the motivation behind this study at the beginning of the Introduction section. Followed by this, interface engineering has been emphasized to show its significant role in catalyst design. Also, we pointed out the importance of studying the property of multi-phase oxides. The corresponding revision has been made on Page 3 of the revised manuscript .

Comment 2: *According to the experiment for the MnCoO_x synthesis, the authors mixed KMnO₄ with cobalt nitrate precursor. However, it is not clear how the authors employed a redox-controlled synthesis method. Moreover, the experimental details for the synthesis are further required. In this regard, the authors need to provide i) which solvent and concentration the author used for the precursors, ii) how the author controlled the synthesis, for example, pH during the synthesis and aging time, and iii) how the authors are sure K is completely removed during the synthesis and post-treatment.*

Reply: We thank the reviewer for pointing this out. A detailed description on the synthesis of MnCoO_x catalysts has been included in the Experimental Section of the revised Supporting Information as well as summarized in Table S10.

Methodology: MnCoO_x catalysts were synthesized by a redox-controlled synthesis method ($\text{Mn}^{7+} + 3\text{Co}^{2+} \rightarrow \text{Mn}^{4+} + 3\text{Co}^{3+}$). In a typical synthesis process, the Mn (VII) solution was prepared by dissolving certain amounts of KMnO₄ into 1000 mL deionized water under magnetic stirring for 30 min at 70 °C. The Co (II) solution was prepared by dissolving specific amounts of Co(NO₃)₂·6H₂O into aqueous solution with certain amounts of potassium citrate under magnetic stirring. Subsequently, the prepared Co precursor solution was added dropwise into the KMnO₄ solution at a specific injection speed to control the reduction process. After completed the injection, the mixed solution was keeping stirring for another 2 h at 80 °C. Then, the mixed solution was maintained under ambient conditions. After aging for a few hours, the black precipitate was collected by filtration, and washed by deionized water and

absolute ethanol three times before drying. After that, the precursor was subjected to an annealing treatment in static air at 350 °C for 2 h at a ramping rate of 1 °C min⁻¹. Finally, the resulting catalysts were washed by 1M NH₄NO₃ solution for 2 h at room temperature under stirring to remove K ions prior to the catalytic tests. The obtained catalysts were denoted as MnCoO_x-z, where z represents the nominal molar ratio of Mn/Co. Here, please see details in Table R4.

Table R4. Synthesis parameters of MnCoO_x catalysts.

Catalysts	Solution I			Solution II		
	KMnO ₄ , g	Solution, mL (H ₂ O)	Co(NO ₃) ₂ ·6H ₂ O, g	Potassium Citrate, g	H ₂ O Solution, mL	Injection speed, mL/min
MnCoO _x -0.1	1	1000	18.25	2.5	200	6.7
MnCoO _x -0.2	1	1000	9.13	1.25	100	3.3
MnCoO _x -0.5	1	1000	3.65	0.50	40	1.33
MnCoO _x -1.0	1	1000	1.8250	0.250	20	0.67
MnCoO _x -2.0	1	1000	0.9125	0.125	10	0.33

Comment 3: *The Raman spectra for the synthetic materials is helpful to understand the structural changes depending on the composition. However, I kindly ask the authors to provide the peak assignment for all the peaks in the spectra, not only stop to the main peak at 670 cm⁻¹ and also more discussion about those peaks according to the compositions.*

Reply: We thank the reviewer for this constructive comment. We agree that Raman spectroscopy is indeed a useful technique to provide structure information of M-O bonds and surface lattice defects. As the reviewer suggested, we have labeled all Raman peaks in the revised manuscript. As shown in **Figure R14**, the band assigned to A_{1g} symmetry at 670 cm⁻¹ is ascribed to the octahedral coordination sites of cobalt (CoO₆), and the band assigned to F_{2g}¹ symmetry at 191 cm⁻¹ belongs to the tetrahedral coordination sites (CoO₄). The Raman bands with medium strength located at 466, 513, and 604 represents E_g, F_{2g}², and F_{2g}³ symmetry.² The information we obtain from

Raman spectra are listed below:

- (1) The Raman spectra of MnCoO_x -0.1 is similar to that of the Co_3O_4 reference. While, the Raman peak of CoO_6 (670 cm^{-1}) gradually shifted to lower wavenumber and merged with the shoulder peak (604 cm^{-1}) to form a broader peak when Mn/Co ratio is above 0.2, implying the weakened vibration of Co-O bonds. Similar phenomenon was also observed in $\text{Ni}_x\text{Co}_{3-x}\text{O}_4$ spinel oxides.⁷ Also, it indicates that the added Mn significantly changed the symmetry of CoO_6 by lattice replacement. The induced coordination environmental change further initiates the occurrence of structural defects and lattice distortion on the developed MnCoO_x , which in turn benefits the formation of oxygen vacancies.
- (2) The peak position of the tetrahedrally coordinated Co sites (CoO_4 : 191 cm^{-1}) was invariant with varied Mn/Co ratios, but the peak intensity of CoO_4 decreased at high Mn/Co ratio due to Mn substitution.
- (3) No active Raman bands belong to Mn-O (as indicated by the blue dash line in Figure Rxx.)⁸⁻¹⁰ were observed in the prepared MnCoO_x catalysts, suggesting that the Mn ions are highly dispersed and/or exist as solid solution in Co_3O_4 spinel.

We have made the corresponding revision on page 5 of the revised manuscript to make it more clearly to the reader.

Figure R14. Raman spectra of MnCoO_x catalysts.

Comment 4: For TOF calculation, the author normalized the reaction rate by metal sites quantified by CO chemisorption. However, if lattice oxygen can be a descriptor for the oxidation reaction and the reaction rate does not rely on the metal sites, the author needs to normalize the rate by the surface or lattice (active) oxygen and then figure out the active site compared to the metal sites.

Reply: We thank the reviewer for this valuable comment. As suggested by the reviewer, we calculated the TOF by normalizing the amounts of surface lattice O* as determined by O₂-TPD analysis. The obtained TOFs were added to Tables S4-S6 in the revised Supporting Information, as summarized in **Table R5**. Noted that, the TOFs calculated by metal sites are comparable to the TOFs calculated by lattice O*, which show similar trends (see **Figure R15**). Here, we would like to clarify that a H₂ reduction was conducted prior to CO chemisorption to determine the number of metal sites by assuming that the surface lattice O is able to be removed during reduction. Therefore, the amount of removed lattice O is about half of the measured metal sites from CO

titration from theoretical point of view.

Table R5. The TOFs of hydrocarbon combustion over MnCoO_x catalysts.

No.	Sample	Amount of surface lattice O* (μmol g ⁻¹) ^a	TOF (h ⁻¹)		
			CH ₄ ^b	C ₂ H ₆ ^c	C ₃ H ₈ ^d
1	MnCoO _x -0.1	249.5	1.12E-04	1.81E-03	6.14E-04
2	MnCoO _x -0.2	321.9	4.28E-04	1.94E-03	1.21E-03
3	MnCoO _x -0.5	385.2	7.17E-04	2.30E-03	1.11E-03
4	MnCoO _x -1.0	239.4	4.19E-04	1.67E-03	5.33E-05
5	MnCoO _x -2.0	199.5	1.39E-04	1.02E-03	3.20E-05
6	Co ₃ O ₄	165.1	3.38E-05	6.08E-04	3.59E-04
7	MnO ₂	264.5	0.00E+00	8.44E-05	0.00E+00

a. The amount of surface lattice O was calculated by multiplying the amount of desorbed O₂ from TPD (refers to Peak II) with two.

b. Turnover frequency (TOF) of CH₄ conversion at 250 °C by normalizing the amounts of surface lattice oxygen (O*).

c. Turnover frequency (TOF) of C₂H₆ conversion at 200 °C by normalizing the amounts of surface lattice oxygen (O*).

d. Turnover frequency (TOF) of C₃H₈ conversion at 150 °C by normalizing the amounts of surface lattice oxygen (O*).

Figure R15. The correlation between TOFs and Mn/Co ratios.

Comment 5: In addition to comment #4, it questions whether the author can provide a quantification of active oxygen for ethane oxidation from O₂-TPD, which only shows the area of the specific peak.

Reply: We thank the reviewer for this valuable suggestion. As mentioned in Comment #4, we provided a quantification of active oxygen from O₂-TPD (see **Table R6**). The amount of surface-active lattice O* is determined by integrating the Peak II of O₂-TPD after curve calibration, as given in Table S3. In the revised manuscript, Figure 2d was updated by replacing the X-axis of “Fitted area of Peak (II)” to “Surface lattice oxygen amount” (**Figure R16**).

Table R6. O₂-TPD of MnCoO_x catalysts and references.

No.	Sample	Total O ₂ desorption amount (μmol g ⁻¹)	Percentage (%) of desorbed O ₂			O ₂ -TPD of Peak II (μmol g ⁻¹) ^a
			Peak (I)	Peak (II)	Peak (III)	
1	MnCoO _x -0.1	226.8	9.03	55.00	35.98	124.7
2	MnCoO _x -0.2	304.2	5.46	52.92	41.62	160.9
3	MnCoO _x -0.5	345.1	3.61	55.81	40.59	192.6
4	MnCoO _x -1.0	310.9	4.72	38.49	56.79	119.7
5	MnCoO _x -2.0	278.2	9.75	35.86	54.39	99.8
6	Co ₃ O ₄	220.2	12.27	37.50	50.24	82.6
7	MnO ₂	263.2	8.19	50.24	41.57	132.2

^a. It represents the amount of surface lattice O.

Figure R16. A correlation of the (a) O₂ desorption amounts of Peak (II) with C₂H₆ oxidation rate; (b) surface lattice O* amount with C₂H₆ oxidation rate.

Comment 6: By considering the apparent activation energy associated with the sensitivity of the materials to the reaction temperature, there seems to be a conflict in the reactivity. For example, MnCoO_{x-0.5} has higher sensitivity to the temperature observed in the light-off ethane profile. Moreover, it shows two different slopes below 20% ethane conversion in the light-off curves. Hence, the authors should describe those different slope changes and then link them to the reactivity.

Reply: We thank the reviewer for this constructive comment. The apparent activation energy (E_a) is an important factor in kinetic studies, which directly reflects the role of a catalyst and its efficiency in certain catalytic reactions. Considering the temperature sensitivity nature of MnCoO_x catalysts, we have recalculated the E_a values based on normalized reaction rates that obtained from a narrow and low ethane conversion range of 5-10% (defined as Region I). As shown in **Figure R17**, the E_a value follows the order of MnCoO_{x-0.5} (81.8 ± 3.2 kJ mol⁻¹) < MnCoO_{x-0.2} (98.3 ± 0.7 kJ mol⁻¹) < MnCoO_{x-0.1} (103.4 ± 1.1 kJ mol⁻¹) < MnCoO_{x-1.0} (108.7 ± 2.2 kJ mol⁻¹) < MnCoO_{x-2.0} (115.6 ± 1.7 kJ mol⁻¹). Noted that, the resulted E_a in Region I is strongly correlated to the reactivity of MnCoO_x catalysts in ethane oxidation. Aside from this, taken MnCoO_{x-0.5} as an example, we have calculated the E_a value at ethane conversion of 10-20% (defined as Region II). Apparently, the E_a value obtained from Region II ($E_a = 95 \pm 8$ kJ mol⁻¹) is higher than that of Region I ($E_a = 81.8 \pm 3.2$ kJ mol⁻¹), suggesting the high sensitivity of MnCoO_{x-0.5} to temperature. Similar phenomenon is also observed over other MnCoO_x catalysts. The corresponding revision has been made on page 8 of the revised manuscript and Figure 2b was updated as well

Figure R17. Catalytic performance of MnCoO_x catalysts for ethane oxidation. (a) Light-off curves of the as-prepared catalysts. (b) corresponding Arrhenius plots.

Comment 7: For oxidation reactivity depending on the chain length of hydrocarbon, it is unclear how the reactivity is limited by the initial H abstraction.

Reply: We thank the reviewer for raising this question. According to the reviewer's suggestion, we modified our expression on page 8 in the revised manuscript to make the statement clearer. It is well-known that the rate of alkane combustion is closely related with the strength of involved C-H bond. Typically, the longer the alkane chain is, the lower the bond dissociation energy (ΔE_{C-H}) is.¹¹ Taken short chain alkanes (C₁-C₃) as an example, we can see that the ΔE_{C-H} follows the trend of CH₄ (465 kJ mol⁻¹) > C₂H₆ (442 kJ mol⁻¹) > C₃H₈ (427 kJ mol⁻¹). The first C-H bond activation of CH₄ is generally regarded as the difficult step due to its high C-H bonding energy, thus it has been regarded as a key factor of governing the activity of methane combustion. In the study by Hu et al.¹², they explored the reaction mechanism of methane combustion over Co₃O₄ by DFT calculations and pointed out that the transition state (TS) of the first C-H activation step is the least stable TS in the whole energy profile. Also, the calculation results showed that the activation barrier of first C-H is the highest by taking the entropy into account, therefore it has been regarded as the rate-determining step in the entire

catalytic cycle of methane combustion on the Co_3O_4 surface. In another study, Tao et al.¹³ stressed out the importance of the first C-H bond dissociation in CH_4 oxidation over the NiCo_2O_4 catalyst. Overall, we can deduce that the length of the alkane chain determines the strength of involved C-H bonds and consequently affects the reactivity of the reaction.¹³ Therefore, the initial H abstraction of short-chain alkanes is often regarded as a key elementary step to assess the reactivity of various catalysts in oxidation reaction.

Meanwhile, a systematic investigation of the potential reaction path was carried out on the (111) plane of the established $\text{MnCo}_2\text{O}_4/\text{MnO}_2$ interfacial model catalyst, which was detailly shown in the response to the Comment #3 of Reviewer 1 (**Figure R5**). The calculated energy diagram of the optimal reaction path over the MnCoO_x model catalysts showed that the energy barrier (ΔE) of dissociating the 1st C-H bond of C_2H_6 is the highest among all the elementary steps, indicating that the 1st dehydrogenation step is the RDS step in ethane oxidation.

Comment 8: *To deconvolute the profile for ethane-TPRS, the authors also need to show MnCo_2O_4 to ensure the peak at 207 °C attributed to $\text{MnO}_2\text{-MnCo}_2\text{O}_4$.*

Reply: We thank the reviewer for this constructive comment. As the reviewer suggested, we have added the C_2H_6 -TPSR result of MnCo_2O_4 reference in the revised Supporting Information (see Figure S18) and improved the relevant expression in the revised manuscript on page 10. As shown in **Figure R18**, there is only one sharp peak appeared at 435 °C on pure MnCo_2O_4 . Through comparing the obtained C_2H_6 -TPSR profile of $\text{MnCoO}_x\text{-0.5}$ catalyst with that of MnO_2 , MnCo_2O_4 , and $\text{MnO}_2/\text{MnCo}_2\text{O}_4$ references, we confirm that the evolved CO_2 peak appeared at 207 °C is attributed to the surface-active oxygen that located at $\text{MnO}_2\text{-MnCo}_2\text{O}_4$ interface.

Figure R18. C₂H₆-TPSR of the MnCo₂O₄ reference.

Comment 9: The authors argued that the higher selectivity C¹⁶O₂ and C^{16/18}O₂ higher than C¹⁸O₂ from TAP indicates the direct participation of lattice O located at MnO₂-MnCo₂O₄ interface. However, it is very hard to find out how the authors reached out this conclusion from the TAP experiment and the oxygen scrambling result. It has questions to prove the interfacial sites for the lattice oxygen in this manuscript.

Reply: We apologize for this unclear statement, which leads to some misunderstandings to the readers. Here, we would like to give more explanations on how we reach this conclusion. **Firstly**, a C₂H₆-TPSR was performed to make qualitative analysis on all the involved O species of the MnCoO_x-0.5 catalyst based on their reducibility. Based on this result, we determine that the O located at MnO₂-MnCo₂O₄ interfacial region is active at low temperature (around 200 °C). **After that**, we took an isotopic labeling experiment at steady-state to study the reaction mechanism. The results show that the surface lattice O played the major role in the redox cycle, especially at lower temperature (around 200 °C), demonstrating that the surface reaction is dominated by the Mars-van Krevelen (MvK) mechanism. **Also**, the designed TAP experiment has confirmed this conclusion. **Moreover**, the experimental data showed that ethane can be completely converted over the MnCoO_x-0.5 catalyst at 220 °C. Through combing the above characterization and experimental results, we ascertain that the lattice O at

MnO₂-MnCo₂O₄ interfaces contributes more on ethane oxidation at lower temperature. We have improved the relevant expressions in the revised manuscript to make this statement clearer (see Page 11).

Reference:

- 1 Rong, S. *et al.* Engineering crystal facet of α -MnO₂ nanowire for highly efficient catalytic oxidation of carcinogenic airborne formaldehyde. *ACS Catal.* **8**, 3435-3446 (2018).
- 2 Zhao, M. *et al.* Roles of surface-active oxygen species on 3DOM cobalt-based spinel catalysts M_xCo_{3-x}O₄ (M = Zn and Ni) for NO_x-assisted soot oxidation. *ACS Catal.* **9**, 7548-7567 (2019).
- 3 Zhang, S. *et al.* WGS catalysis and in situ studies of CoO_{1-x}, PtCo_n/Co₃O₄, and Pt_mCo_m/CoO_{1-x} nanorod catalysts. *J. Am. Chem. Soc.* **135**, 8283-8293 (2013).
- 4 Fu, C. *et al.* One-step calcination-free synthesis of multicomponent spinel assembled microspheres for high-performance anodes of Li-ion batteries: a case study of MnCo₂O₄. *ACS Appl. Mater. Inter.* **6**, 2439-2449 (2014).
- 5 Xie, S. *et al.* Insights into the active sites of ordered mesoporous cobalt oxide catalysts for the total oxidation of o-xylene. *J. Catal.* **352**, 282-292 (2017).
- 6 Wang, T. *et al.* Facile synthesis of palladium incorporated NiCo₂O₄ spinel for low temperature methane combustion: Activate lattice oxygen to promote activity. *J. Catal.* **404**, 400-410 (2021).
- 7 Liu, S. *et al.* Engineering morphology and Ni substitution of Ni_xCo_{3-x}O₄ spinel oxides to promote catalytic combustion of ethane: Elucidating the influence of oxygen defects. *ACS Catal.* **13**, 4683-4699, (2023).
- 8 Barai, H. R. *et al.* Improved electrochemical properties of highly porous amorphous manganese oxide nanoparticles with crystalline edges for superior supercapacitors. *J. Ind. Eng. Chem.* **56**, 212-224 (2017).
- 9 Jayashree, M. *et al.* Ultrafine MnO₂/graphene based hybrid nanoframeworks as high-performance flexible electrode for energy storage applications. *J. Mater. Sci. Mater. Electron.* **31**, 6910-6918 (2020).
- 10 Soldatova, A. V. *et al.* Biogenic and synthetic MnO₂ nanoparticles: Size and growth probed with absorption and Raman spectroscopies and dynamic light scattering. *Environ. Sci. Technol.* **53**, 4185-4197, (2019).
- 11 Deshlahra, P. *et al.* Reactivity and selectivity descriptors for the activation of C-H bonds in hydrocarbons and oxygenates on metal oxides. *J. Phys. Chem. C* **120**, 16741-16760, (2016).
- 12 Hu, W. *et al.* Origin of efficient catalytic combustion of methane over Co₃O₄(110): Active low-coordination lattice oxygen and cooperation of multiple active sites. *ACS Catal.* **6**, 5508-5519, (2016).
- 13 Tao, F. F. *et al.* Understanding complete oxidation of methane on spinel oxides at a molecular level. *Nature Commun.* **6**, 1-10 (2015).

REVIEWER COMMENTS

Reviewer #1 (Remarks to the Author):

I am satisfied with the authors' answers to my critical comments. The manuscript was revised according to the comments by the referee and is acceptable to this journal.

Reviewer #2 (Remarks to the Author):

In this work, the authors developed the MnCoO_x catalysts by chemical reduction for ethane oxidation, and create the MnO₂-Mn_xCo_{3-x}O₄ interface through changing the Mn/Co ratio, it exhibits an excellent activity and stability when Mn/Co ratio is 0.5. They also revealed the role of the interface between MnO₂-Mn_xCo_{3-x}O₄ components during the process of the ethane oxidation, in which the C₂H₆ tends to be adsorbed on the interfacial Co sites and subsequently break the CH bonds on the reactive lattice O of MnO₂ layer. This work is novel and useful to readers in this field, and it is well organized and written. In addition, the author has addressed all my comments. In my opinion, it is ready for publication.

(1)The introduction in the present work should be improved. It is suggested to be focused much more on the role of MnCoO₄ and the interface engineering.

(2)Since the structure of MnO₂-Mn_xCo_{3-x}O₄ is a heterogeneous, the DFT model, which is agree with experimental data, is needed to be further explained.

Reviewer #3 (Remarks to the Author):

I would thank the authors for their careful revision. The authors have addressed the questions raised by the reviewers including me in the revised manuscript to Nat. Commun. Therefore, I recommend the revised manuscript to be accepted directly.

RESPONSE TO REVIEWERS' COMMENTS

In this response letter, comments from the reviewers are summarized in black typeface with the original comments quoted in *italic*, and our responses are in blue typeface. All major changes have been highlighted in yellow in the main text. We have attached a clean version of the revised manuscript and a file with tracking changes.

A point-to-point response to the comments from three reviewers

Reviewer #1

I am satisfied with the authors' answers to my critical comments. The manuscript was revised according to the comments by the referee and is acceptable to this journal.

RESPONSE:

We would like to thank the reviewer again for your valuable comments and suggestions to help us improve the quality of this manuscript.

Reviewer #2:

In this work, the authors developed the $MnCoO_x$ catalysts by chemical reduction for ethane oxidation, and create the $MnO_2-Mn_xCo_{3-x}O_4$ interface through changing the Mn/Co ratio, it exhibits an excellent activity and stability when Mn/Co ratio is 0.5. They also revealed the role of the interface between $MnO_2-Mn_xCo_{3-x}O_4$ components during the process of the ethane oxidation, in which the C_2H_6 tends to be adsorbed on the interfacial Co sites and subsequently break the CH bonds on the reactive lattice O of MnO_2 layer. This work is novel and useful to readers in this field, and it is well organized and written. In addition, the author has addressed all my comments. In my opinion, it is ready for publication.

(1) The introduction in the present work should be improved. It is suggested to be focused much more on the role of $MnCoO_4$ and the interface engineering.

(2) Since the structure of $MnO_2-Mn_xCo_{3-x}O_4$ is a heterogeneous, the DFT model, which is agree with experimental data, is needed to be further explained.

RESPONSE:

We thank the reviewer for the overall positive reviews on our work. We also thank the reviewer for the constructive comments. According to the reviewer's suggestions, we have made further revision in the **Introduction** part to highlight the important role of spinel supports as well as the interface tuning strategies. Several recent publications have been added in the revised manuscript. For example, Done et al.¹ stressed out the important role of metal doping on alternating the electron configuration of metal ions in AB_2O_4 spinel, which in turn affects the adsorption strength of reactants. This given example here has emphasized the tunable nature of spinel oxides on varying the local electron environment of involved metal species and further affecting the adsorption strength of reactants. Shan et al.² adopted the acid-etching approach to create $MnO_2-CoMn_2O_4$ interfacial system and unveiled that the lattice O that located at

interfacial sites was activated due to the weakened Mn-O bonds as well as the altered coordination environments of O atoms. Also, Ren et al.³ discovered that the concentration of oxygen vacancies of CoMn_2O_4 spinel significantly increased after HNO_3 treatment, therefore generating more active surface O species during O_2 activation. Zhang et al.⁴ constructed $\text{AgO}/\text{CeSnO}_x$ tandem catalysts and studied the synergistic effects between AgO and CeSnO_x dual sites for selective oxidation of NH_3 . It is noticed that the electrons on CeSnO_x support were more easily transferred to AgO NPs, which accelerates the oxidation activity of AgO and the reduction performance of CeSnO_x support, resulting in a good match between NH_3 oxidation and NO_x reduction. Also, the strong interaction between different metal oxides affects the dispersion and crystallinity of active centers.^{5,6}

In response to the suggestion on the DFT models, more detailed explanations have been provided in the *Surface Mechanism* section to illustrate how we built the MnO_2 - MnCo_2O_4 interfacial model. Firstly, we constructed the MnCo_2O_4 crystal structure by replacing part of the Co atoms of cubic Co_3O_4 with Mn. As shown in **Figure R1** (supplementary Fig. 29(a)), the Type (II) model was found to be the most stable structure by substituting octahedral Co^{3+} with Mn^{3+} , as demonstrated by the lowest relative energy per Mn atom in the proposed MnCo_2O_4 models from DFT calculations. The obtained lattice parameter of MnCo_2O_4 is enlarged from 8.07 to 8.14 Å, which is consistent with the XRD results. Followed by this, the crystal facets of MnO_2 and MnCo_2O_4 were chosen based on the HRTEM results. After analyzing the termination stability of MnO_2 and MnCo_2O_4 (supplementary Fig. S30 (a)), the optimized MnO_2 - MnCo_2O_4 interfacial models were established by taking MnCo_2O_4 -111-A as the underlying substrate and intercepting a structural unit from MnO_2 -111-C, MnO_2 -110-B, and MnO_2 -101-B as the upper cluster (named as $\text{MnCo}_2\text{O}_4/\text{MnO}_2$ -111-C, $\text{MnCo}_2\text{O}_4/\text{MnO}_2$ -110-B, and $\text{MnCo}_2\text{O}_4/\text{MnO}_2$ -101-B, respectively, see details in Supplementary Figs. 30-31). We have revised the manuscript accordingly by incorporating all of Reviewer 2's comments, please see the yellow highlighted areas in the revised manuscript.

Figure R1. Relative energy per Mn atom in the proposed MnCo₂O₄ models.

Reviewer #3:

I would thank the authors for their careful revision. The authors have addressed the questions raised by the reviewers including me in the revised manuscript to *Nat. Commun.* Therefore, I recommend the revised manuscript to be accepted directly.

RESPONSE:

We would like to thank the reviewer again for your valuable comments and suggestions to help us improve the quality of this manuscript.

Reference:

- 1 Dong, C. *et al.* Local electron environment regulation of spinel CoMn₂O₄ induced effective reactant adsorption and transformation of lattice oxygen for toluene oxidation. *Environ. Sci. Technol.* **57**, 21888-21897 (2023).
- 2 Shan, C. *et al.* Acid etching-induced in situ growth of λ-MnO₂ over CoMn spinel for low-temperature volatile organic compound oxidation. *Environ. Sci. Technol.* **56**, 10381-10390 (2022).

- 3 Ren, Y. *et al.*, A. Acid-etched spinel CoMn_2O_4 with highly active surface lattice oxygen species for significant improvement of catalytic performance of VOCs oxidation. *Chem. Eng. J.* **463**, 142316 (2023).
- 4 Zhang, Y. *et al.* Elimination of NH_3 by interfacial charge transfer over the Ag/CeSnO_x tandem catalyst. *ACS Catal.* **13**, 1449-1461 (2023).
- 5 Song, J. *et al.* LDH derived MgAl_2O_4 spinel supported Pd catalyst for the low-temperature methane combustion: Roles of interaction between spinel and PdO. *Appl. Catal. A: Gen.* **621**, 118211 (2021).
- 6 Shan, C. *et al.* Recent advances of VOCs catalytic oxidation over spinel oxides: Catalyst design and reaction mechanism. *Environ. Sci. Technol.* **57**, 9495-9514 (2023).